# High-resolution single-photon imaging with physics-informed deep learning

Liheng Bian[1,2,5] ✉, Haoze Song[1,5], Lintao Peng[1], Xuyang Chang ®[1], Xi Yang[3], Roarke Horstmeyer[3], Lin Ye[4], Chunli Zhu[1], Tong Qin[1], Dezhi Zheng ®[1,2] & Jun Zhang[1] ✉

High-resolution single-photon imaging remains a big challenge due to the complex hardware manufacturing craft and noise disturbances. Here, we introduce deep learning into SPAD, enabling super-resolution single-photon imaging with enhancement of bit depth and imaging quality. We first studied the complex photon flow model of SPAD electronics to accurately characterize multiple physical noise sources, and collected a real SPAD image dataset (64 × 32 pixels, 90 scenes, 10 different bit depths, 3 different illumination flux, 2790 images in total) to calibrate noise model parameters. With this physical noise model, we synthesized a large-scale realistic single-photon image dataset (image pairs of 5 different resolutions with maximum megapixels, 17250 scenes, 10 different bit depths, 3 different illumination flux, 2.6 million images in total) for subsequent network training. To tackle the severe super-resolution challenge of SPAD inputs with low bit depth, low resolution, and heavy noise, we further built a deep transformer network with a content-adaptive self-attention mechanism and gated fusion modules, which can dig global contextual features to remove multi-source noise and extract full-frequency details. We applied the technique in a series of experiments including microfluidic inspection, Fourier ptychography, and high-speed imaging. The experiments validate the technique's state-of-the-art super-resolution SPAD imaging performance.

Single-photon avalanche diode (SPAD) array has received wide attention due to its excellent single-photon sensitivity[1-4]. Such a single-photon imaging sensor has been widely applied in various fields such as fluorescence lifetime imaging[5], fluorescence fluctuation spectroscopy[6], time-off-light imaging[7-9], quantum communication and computing[10,11], and so on[12,13]. Compared with EMCCD and sCMOS cameras that also maintain high detection sensitivity, SPAD arrays acquire photon-level light signals at a low-noise level, and perform direct photon-digital conversion that can effectively eliminate readout noise and enhance readout speed[14].

While early SPAD arrays were limited in imaging resolution due to low fill factors, which depend on a large guard ring of each pixel to prevent premature edge breakdown during electron avalanche[15], recent advance has exhibited higher fill factors approaching 100%[16]. Nevertheless, compared with the well-established manufacture craft of CMOS using which Kitamura et al. have reported a 33 megapixel standard CMOS imaging sensor (7680 × 4320 pixels) in 2012[17], the array size of SPAD in academic literature was only 400 × 400 in 2019[18] and 1024 × 1000 in 2020[15]. Moreover, the commercial SPAD products available in the market maintain only thousand pixels[19], and cost more

[1]MIIT Key Laboratory of Complex-field Intelligent Sensing, Beijing Institute of Technology, Beijing 100081, China. [2]Yangtze Delta Region Academy of Beijing Institute of Technology (Jiaxing), Jiaxing 314019, China. [3]Department of Biomedical Engineering, Duke University, Durham, NC 27708, USA. [4]School of Materials Science and Engineering, Beijing Institute of Technology, Beijing 100081, China. [5]These authors contributed equally: Liheng Bian, Haoze Song. ✉e-mail: bian@bit.edu.cn; zhjun@bit.edu.cn

than twenty thousand euros. In this regard, the average pixel cost of SPAD is 8 orders of magnitude higher than that of a commercial CMOS. To improve SPAD imaging resolution, Sun et al. reported an optically coded super-resolution technique[20] with a high-SNR input with 5.2 ms integration time, which is several orders higher than practical single-photon imaging applications with sub-nanosecond exposure time[7–9]. The underlying limitation originates from the employed single-source Poisson noise model[21,22], which deviates from complex real SPAD noise containing a variety of different-model sources such as crosstalk, dark count rate, and so on[1] (as shown in Fig. 1a). Mora-Martín et al. presented a technique that utilizes synthetic SPAD depth sequences to train a 3D convolutional neural network (CNN) for denoising and upscaling (4×)[23]. However, the technique also falls into the single-source based Poisson and Gaussian noise statistics that lead to degraded imaging quality without considering such multiple noise sources (as validated in the following experiments shown in Fig. 2b).

In the era of deep learning, another obstacle preventing super-resolution SPAD imaging is the lack of datasets containing image pairs of low-resolution single-photon acquisitions and high-resolution ground truth. Although one can accumulate multiple subframes to generate noise-free images, its pixel resolution is still low and far from meeting the Nyquist sampling requirement. One way to acquire high-resolution images is placing another CMOS or CCD camera to take images of the same target, which however introduces additional laboursome registration workload[24]. Besides the extremely low resolution and complex noise model, the single-photon image enhancement task faces a more severe challenge due to its low bit depth that is different from the conventional image enhancement tasks on CMOS or CCD acquired images[25–28]. In such a case, the local features employed by either the conventional optimization techniques[29,30] or the convolutional neural network techniques[26,28] are limited to predicting neighboring SPAD pixels that may vary drastically (as seen below).

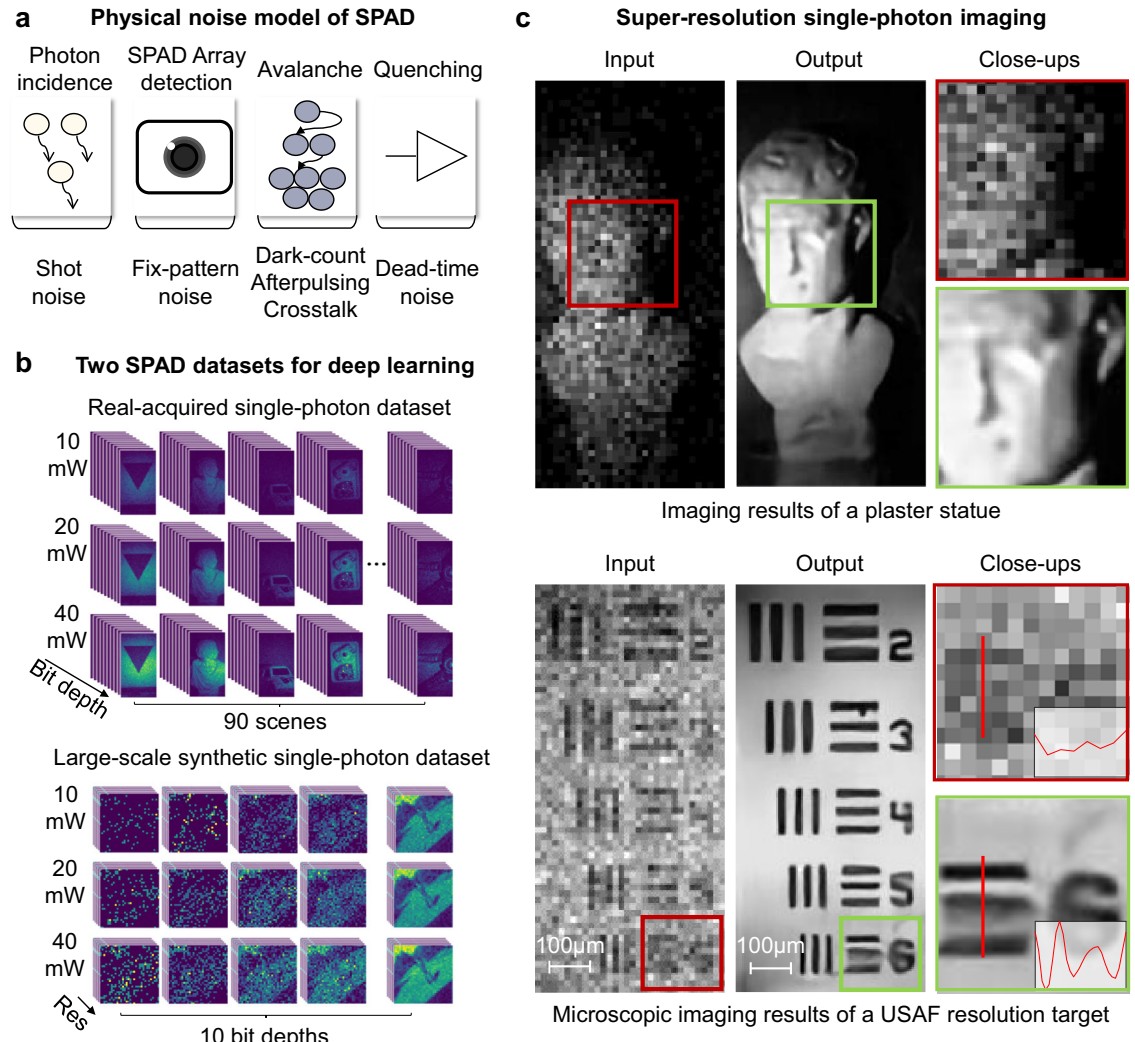

**Fig. 1 | Illustration of the reported large-scale single-photon imaging technique. a** The multi-source physical noise model of Single-photon avalanche diode (SPAD) arrays, which consists of shot noise from photon incidence, fixed-pattern noise from SPAD array's photon absorption, dark count rate, afterpulsing and crosstalk noise from blind electron avalanche, and deadtime noise from the quenching circuit. **b** The visualization of two collected SPAD image datasets, one of which contains acquired single-photon images (64 × 32 pixels, 90 scenes) under 10 different bit depth and 3 different illumination fluxes. This dataset was applied to calibrate noise model parameters. The other dataset was digitally synthesized that contains 2.6 million images of 17,250 scenes (image pairs of 5 different resolutions with maximum megapixels, 10 different bit depth, 3 different illumination flux). The large-scale synthetic data was applied to train neural networks for single-photon enhancement. **c** The exemplar super-resolution SPAD imaging results including a plaster statue and a United States Air Force (USAF) resolution target. The direct acquired SPAD images (64 × 32 pixels, 6 bits) were input into the enhancing neural network, which produces high-fidelity super-resolution images (256 × 128 pixels) with fine details and smooth background.

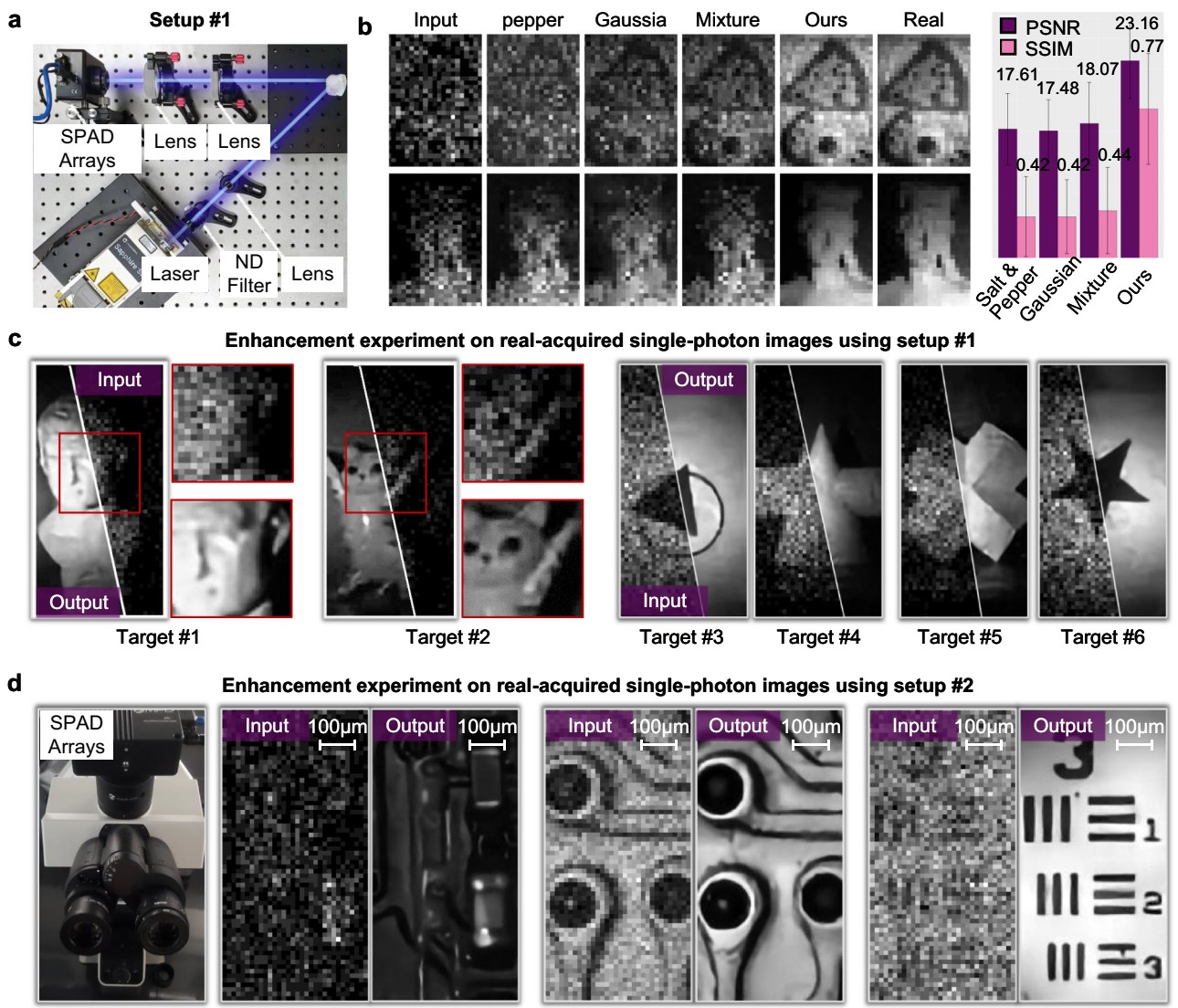

**Fig. 2 | Experiment results of large-scale single-photon imaging. a** The optical setup to acquire single-photon images of various targets. **b** The comparison of enhanced single-photon imaging results using different noise models. We used different noise models to train the same neural network, and input acquired data into the different trained models for enhancement. The ground-truth real image for reference was accumulated using 60,000 binary single-photon images. **c** The enhanced single-photon images of different macro targets (including a plaster model, a cat toy and different printed shapes) using the reported technique (with 6-bit inputs). **d** The single-photon microscopy setup and corresponding enhancing results using the reported technique. The setup was built by mounting the SPAD camera on an off-the-shelf microscope, to acquire single-photon images of microscopic samples in 6 bits. The enhanced single-photon results of a printed circuit board and a UASF resolution target are presented on the right side.

In this work, to tackle the great challenge of high-fidelity super-resolution SPAD imaging with low bit depth, low resolution and heavy noise in photon-limited scenarios, we first established a real-world physical noise model of SPAD arrays. As shown in Fig. 1a, the real physical noise sources consist of shot noise from photon incidence, fixed-pattern noise from SPAD array's photon absorption, dark count rate, afterpulsing and crosstalk noise from blind electron avalanche, and deadtime noise from the quenching circuit. To calibrate the parameters of such a complex multi-source noise model, we collected a real-shot SPAD image dataset containing 2790 images in total, each with $64 \times 32$ pixels. Among these images, there are 90 scenes, each with 10 different bit depths and 3 different illumination fluxes for studying different application conditions, as shown in Fig. 1b. With the calibrated physical noise model under different illumination and acquisition settings, we further employed off-the-shelf public high-resolution images (collected from the PASCAL VOC2007[31] and

VOC2012[32] datasets) to digitally synthesize a large-scale realistic single-photon image dataset containing 2.6 million image pairs.

Driven by the single-photon image dataset, we designed a gated fusion transformer network for single-photon super-resolution enhancement. The transformer framework[33,34] has recently attracted increasing attention and produced an impressive performance on multiple vision tasks[34–36]. As presented below, the reported network mainly consists of three modules, including the shallow feature extraction module, the deep feature fusion module and the image reconstruction module. The framework can gradually extract multi-level frequency features of input images and perform adaptive weighted fusion to reconstruct high-quality images. The gated fusion transformer network was trained using the above large-scale single-photon image dataset and tested on various SPAD images. We built four experiment setups to acquire various macroscopic and microscopic images, two of which acquired target images for dataset

collection and direct enhancement validation, and another two were applied for microfluidic inspection and Fourier ptychograpic imaging. A series of experiments validate the technique's state-of-the-art super-resolution single-photon imaging performance.

## Results

### Collected dataset for noise model evaluation

We first built an optical setup to acquire SPAD images of various targets, as shown in Fig. 2a. The setup was built on an optical table in a darkroom, which consists of a 488 nm laser (Coherent Sapphire SF 488-100), a neutral density filter, a set of lenses and a SPAD array camera (MPD-SPC3). The target is illuminated by the laser, and the reflected light is focused to the SPAD array. Using the setup, we first collected a real-acquired SPAD image dataset containing 90 targets with 9 classes (element, geometry, sculpture, rock, plant, animal, car, food, and lab tool), as illustrated in Fig. 1b. For each target, we set the laser power being 10 mW, 20 mW, and 40 mW, respectively. Under each illumination, the integration time of single-photon detection is $0.02\mu s$, and we acquired 1024 images for each target to synthesize multiple bit depths ranging from 1 to 10. As such, the frame time is $20\mu s$. The captured images are at the resolution of $32 \times 64$ pixels. These images were applied to calibrate the multi-source noise model parameters in Eq. (1). More parameter details are referred in Supplementary Note 1.

To validate the fidelity of the reported multi-source noise model, we used the synthesized images of different noise models to train different neural networks under the same transformer framework (without super resolution yet), and compared their enhanced results with the reference ground-truth (GT) image composed of 60,000 SPAD subframes. As shown in Fig. 2b, the enhanced images using the reported multi-source model maintain the highest fidelity and fine details compared to ground truth, while the other models produce various aberrations with remaining noise that is hard to remove. Besides, we note that the GT image exhibits a smooth background and fine target details, which alleviates the concern of coherent speckle noise coming from laser illumination. The quantitative comparison using the peak signal-to-noise ratio (PSNR) and structural similarity (SSIM) as metrics are presented on the right side of Fig. 2b, which indicates that our technique produces more than 5 dB higher PSNR and 0.3 higher SSIM than the other models. Such superiority validates that the reported multi-source physical noise model and calibration strategy enable an accurate description of SPAD's photon flow. The details of collected imaging datasets are referred in the Supplementary Note 2.

### Large-scale dataset for super-resolution SPAD imaging

Despite the above collected single-photon image dataset, further super-resolution enhancement cannot be achieved due to the lack of high-fidelity and high-resolution ground truth pairs. To tackle such a big challenge, we further synthesized a large-scale single-photon image dataset using the public copyright-free Pascal voc2007 and voc2012 images (24 bits). Specifically, the high-resolution images were cropped to $512 \times 512$ pixels and downsampled to different scales of 2, 4, 8, and 16 in each dimension. Then, to simulate acquired SPAD images, measurement noise of different illumination and bit depth conditions were added to the $32 \times 32$ images using the above calibrated noise parameters. In this way, there produce 17250 image pairs of low-resolution SPAD images ($32 \times 32$ pixels, 4 different bit depths and 4 different illumination flux) and corresponding high-resolution ground truth. The dataset structure is demonstrated in Fig. 1b.

We trained the reported gated fusion transformer neural network (detailed in the Methods section) on the synthesized image dataset under different bit depths, illumination conditions, and super-resolution scales, and tested the network and state-of-the-art enhancing techniques on the test dataset including 505 synthetic noise images. Each test image is composed of different subframes ranging from 16 to 1024 (corresponding to the bit depth ranging from 4 to 10), where each pixel is 1 or 0 indicating whether a photon is detected. The competing methods are classified into two categories, including the popular optimization-based denoising techniques (BM3D and VST) together with subsequent bicubic interpolation ("BM3D+bicubic" and "VST+bicubic"), and the state-of-the-art neural networks (U-net, Memnet and SwinIR). These networks were also trained on our synthesized dataset to converge for a fair comparison.

We list a quantitative comparison of peak signal-to-noise ratio (PSNR) and structural similarity metric (SSIM) in Table 1. We can see that the reported technique achieves the best performance for all the different input settings. Especially, compared to the conventional enhancing methods such as BM3D, our method produces more than 5 dB higher PSNR and more than 0.2 higher SSIM. Compared to the competing neural networks, the reported transformer network maintains as much as 2 dB superiority on PSNR. The results further validate

**Table 1 | Quantitative enhancement comparison among different enhancing methods and networks under different bit depth (subframe), illumination flux (laser power), and super-resolution scale**

| Subframe | Laser power/mW | SR scale | Metric | BM3D +bicubic | VST +bicubic | U-net | Memmnet | SwinIR | Ours |
|----------|----------------|----------|--------|---------------|--------------|-------|---------|--------|------|
| 512 | 10 | 2 | PSNR | 18.50 | 16.38 | 19.87 | 20.34 | 21.26 | **23.02**[a] |
|  |  |  | SSIM | 0.52 | 0.35 | 0.59 | 0.64 | 0.68 | **0.79** |
|  |  | 4 | PSNR | 19.21 | 18.10 | 20.06 | 20.86 | 21.47 | **22.52** |
|  |  |  | SSIM | 0.49 | 0.48 | 0.62 | 0.66 | 0.68 | **0.75** |
|  | 40 | 2 | PSNR | 18.81 | 16.10 | 20.71 | 21.23 | 22.04 | **24.03** |
|  |  |  | SSIM | 0.59 | 0.36 | 0.62 | 0.69 | 0.71 | **0.83** |
|  |  | 4 | PSNR | 19.61 | 18.11 | 20.56 | 2103 | 21.75 | **23.58** |
|  |  |  | SSIM | 0.56 | 0.48 | 0.61 | 0.67 | 0.72 | **0.79** |
| 1024 | 10 | 2 | PSNR | 19.61 | 16.68 | 20.44 | 21.01 | 21.76 | **23.59** |
|  |  |  | SSIM | 0.58 | 0.36 | 0.63 | 0.67 | 0.72 | **0.82** |
|  |  | 4 | PSNR | 20.55 | 18.73 | 20.24 | 20.87 | 21.32 | **23.28** |
|  |  |  | SSIM | 0.56 | 0.48 | 0.61 | 0.67 | 0.72 | **0.79** |
|  | 40 | 2 | PSNR | 18.58 | 15.91 | 21.59 | 22.10 | 22.9 | **24.55** |
|  |  |  | SSIM | 0.63 | 0.36 | 0.68 | 0.70 | 0.72 | **0.84** |
|  |  | 4 | PSNR | 19.35 | 17.83 | 21.33 | 21.97 | 22.45 | **24.07** |
|  |  |  | SSIM | 0.61 | 0.49 | 0.65 | 0.71 | 0.75 | **0.82** |

[a]Bold number indicates the highest metric in the comparison.

**Table 2 | Network evaluation on different bit depth, with the illumination flux fixed to be 40 mW and the super-resolution scale fixed to be 4**

| Subframes | index | BM3D +bicubic | VST +bicubic | U-net | Memmnet | Swinir | Ours |
|---|---|---|---|---|---|---|---|
| 16 | PSNR | 11.85 | 12.82 | 19.94 | 20.23 | 20.48 | **22.36**[a] |
|  | SSIM | 0.18 | 0.42 | 0.56 | 0.61 | 0.64 | **0.71** |
| 32 | PSNR | 13.74 | 14.88 | 19.89 | 20.49 | 20.78 | **22.84** |
|  | SSIM | 0.31 | 0.46 | 0.58 | 0.63 | 0.66 | **0.73** |
| 64 | PSNR | 15.60 | 16.64 | 20.03 | 20.52 | 20.91 | **23.21** |
|  | SSIM | 0.30 | 0.48 | 0.59 | 0.65 | 0.69 | **0.73** |
| 128 | PSNR | 17.01 | 17.58 | 20.15 | 20.63 | 21.37 | **23.43** |
|  | SSIM | 0.42 | 0.5 | 0.6 | 0.64 | 0.71 | **0.76** |
| 256 | PSNR | 17.89 | 17.70 | 20.49 | 20.89 | 21.67 | **23.49** |
|  | SSIM | 0.47 | 0.51 | 0.62 | 0.66 | 0.7 | **0.77** |
| 512 | PSNR | 19.61 | 18.11 | 20.56 | 21.03 | 21.75 | **23.58** |
|  | SSIM | 0.56 | 0.48 | 0.62 | 0.67 | 0.72 | **0.79** |
| 1024 | PSNR | 19.35 | 17.83 | 21.33 | 21.97 | 22.45 | **24.07** |
|  | SSIM | 0.61 | 0.49 | 0.65 | 0.71 | 0.75 | **0.82** |

[a]Bold number indicates the highest metric in the comparison.

the effectiveness of our technique in attenuating measurement noise and retrieving fine details.

Further, we fixed the illumination flux to be 40 mW, and the super-resolution scale to be 4, and studied these method's enhancement performance under different bit depths ranging from extremely-low 16 subframes to 1024 frames. The quantitative comparison results are presented in Table 2, from which we can see that the reported technique maintains excellent robustness to different subframes. The PSNR metric degrades less than 2 as the subframe number decreases from 1024 to 16, in which case the conventional BM3D+bicubic technique obtains PSNR degradation higher than 7.

### Enhancement on real-acquired data

We applied the trained neural network on real-acquired data for experiment validation. The acquired images were input into the enhancing network with the super-resolution scale being 4 (output 128 × 256 pixels). The targets include a plaster model, a cat toy and different printed shapes. The enhanced results are presented in Fig. 2c. We can see that the reported technique produces high-fidelity super-resolution results, and maintains strong generalization on different targets. Specifically for the plaster model (target #1), as shown in the close-ups, the nose and mouth details are completely missing in the input image, while they are clearly retrieved in the enhanced results. The similar observation also exists for the cat toy (target #2). The comparison results of the different shape targets reveal that the reported technique is able to produce a smooth background while reconstructing fine structure details. We provide more enhancement results on real-acquired images in Supplementary Note 5. The experiment settings and specifications are provided in Supplementary Note 11 for easy reference.

To further validate the enhancing ability on microscopic imaging, we built the second microscopy setup by mounting the SPAD array camera (MPD-SPC3) to a commercial microscope (Olympus BX53). The hardware integration time is still set as $0.02\mu s$, ensuring that each frame's maximum pixel value is 1 to generate a 1-bit image. This allows us to synthesize input images at different bit depths. The numerical aperture of the microscope objective is 0.25. The micro targets include a circuit board and a United States Air Force (USAF) resolution test target (R3L3S1N, Thorlabs). The enhanced results are presented in Fig. 2d. For the circuit board, we can clearly observe the resistance component and connecting wires in the enhanced super-resolution images,

while such details are buried in heavy noise in the low-resolution and low-bit-depth acquired images. For the resolution test target, it is hard to even discriminate element 1 of group 3 for direct SPAD detection. In comparison, the enhanced resolution achieves element 6 of group 3 (Fig. 1c), with a smooth background and clear details.

### Experiment on microfluidic inspection

Microfluidic technique manipulates and processes small amounts of fluids using mini channels of micrometer width[37]. Inspection of microfluidic chips is important to monitor fluid flow. We built a microfluidic inspection setup using SPAD, as shown in Fig. 3a. A custom-made microfluidic device was constructed using two syringe pumps (BT01100, Longer Pump Limited Company) and a microfluidic chip (Wenhao Microfluidic Technology Limited Company). The chip and pumps were connected by polytetrafluoroethene (PTFE) tube with an inner diameter of 0.6 mm and outer diameter of 1.6 mm. The aqueous lemon yellow solution (2 mg/mL, Meryer Chemical Reagent Company) was used as the disperse phase, and the liquid paraffin (Macklin Chemical Reagent Company) was used as the continuous phase. The former was pumped into the middle channel of the chip, and the latter flowed into the chip from the side channels. After these two kinds of liquid met in the middle channel, the lemon yellow solution was sheared by the continuous phase and formed micro-droplets in the channel. The flow rate of the continuous and dispersed phase was 1 mL/h and 0.1 mL/h, respectively. The micro-droplets formed by lemon yellow solution were driven forward by the flow of the paraffin.

The micro-droplet formation process was recorded using the SPAD camera, as presented in the upper row in Fig. 3b. The images were accumulated using 1024 binary single-photon images, corresponding to 10-bit depth. However, due to the extremely low resolution and heavy noise, it is hard to reveal the fluid flow process or even the micro channel structure from the acquired single-photon images. The channel boundaries are barely visible in the measurements. The closeups at the left bottom of each image can barely render the micro-droplet structure but only a clutter of pixels. In comparison, by applying the reported enhancement technique on the acquired single-photon images, the enhanced images displayed in the bottom row in Fig. 3b can reveal much more details of the micro-droplets and micro channels. The width evolutionary process of microdroplets can be conveniently calculated from the super-resolution images, which is

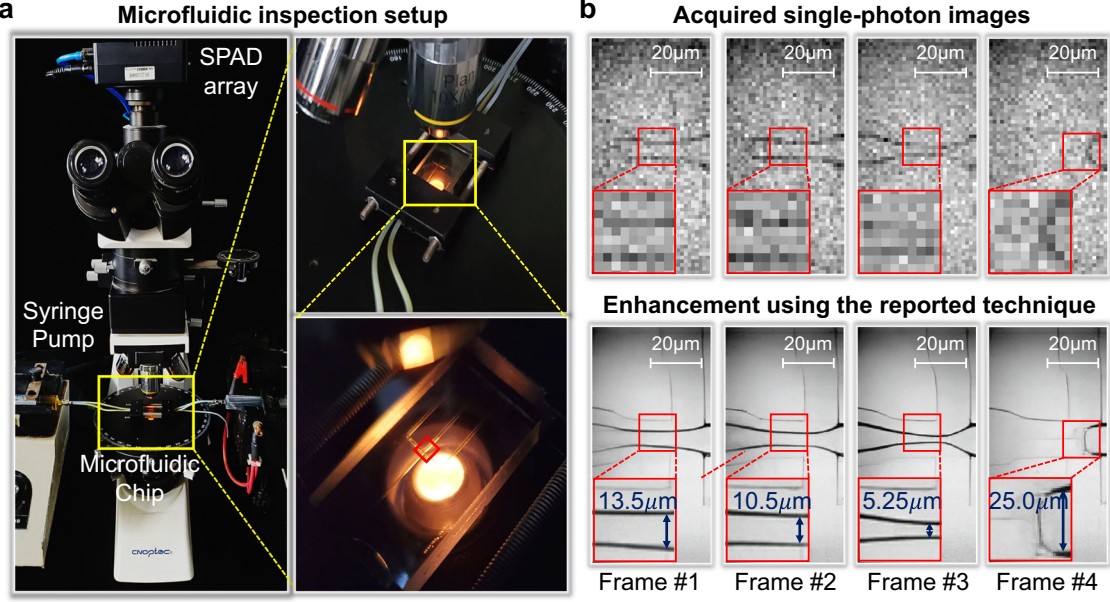

**Fig. 3 | Microfluidic inspection experiment using single-photon detection. a** The microfluidic inspection setup to monitor fluid flow in microfluidic chip. **b** The acquired single-photon images at different time in 8 bits and corresponding enhancement results.

impossible for the low-resolution single-photon images where the pixel size is as large as $30\mu m$.

Using the microfluidic inspection setup, we can clearly study fluid flow and micro-droplet change process from frame sequence. As shown in the four frames in Fig. 3b, with the gradual increase of the continuous phase shear force in the microfluidic channel, the dispersed phase fluid gradually becomes thinner until it fractures. The phenomenon originates from that in a microfluidic chip with a flow-convergence structure, two immiscible liquids coaxially enter different microchannels of the chip. Among them, the continuous phase liquid flows in from the microchannels on both sides of the dispersed phase liquid, and produces an interflow extrusion effect on the dispersed phase liquid. At the junction of the channels, the dispersed phase is sheared by the continuous phase and extends into a "finger-like" or "nozzle-like" shape. Afterward, under the action of the liquid flow field, the dispersed phase is squeezed or sheared, breaking into uniform droplets. More microfluidic inspection experiment results are provided as Supplementary Note 6.

### Experiment on Fourier ptychographic microscopy

Fourier ptychographic microscopy (FPM) is a synthetic coherent imaging technique, providing high-throughput amplitude and quantitative phase images[38,39]. It works by acquiring multiple images under different angles of illumination, and inputting these images into phase retrieval algorithms for simultaneous amplitude and phase reconstruction. In our recent work[40], we have successfully implemented FPM with SPAD array, realizing coherent imaging with higher resolution and larger dynamic range with single-photon inputs. The experiment setup is presented in Fig. 4a, where a 0.1 NA Olympus plan achromatic objective with a 600 mm focal length tube lens was applied for imaging, providing ×18.5 magnification for Nyquist sampling. An Adafruit P4 Light Emitting Diode (LED) array was placed 84 mm from the sample to provide different angles of illumination, among which we used 9 × 9 632 nm LEDs. The illumination NA was 0.18.

The exemplar acquired images of an onion cell sample, a blood cell sample (simulated data), a USAF resolution test target and a plant sample (real-acquired data) are presented in the left column of Fig. 4b, c, respectively. The acquired images were all taken under vertical illumination, with the bit depth being 9 (accumulated by 512 binary single-photon images). We first applied the standard FPM reconstruction algorithm[40] to stitch the acquired SPAD images of different illumination angles together, and the FPM reconstruction results are shown in the middle column. We can see that although FPM effectively enhances imaging resolution, there still exists obvious noise and aberration in the reconstruction that is introduced by heavy SPAD noise. Then, we applied the reported technique on the FPM reconstruction, and the enhanced results are presented in the last columns. We can see that in the improved results, the background is smoothed, while the super-resolution details are further enhanced. The experiment further validates the superior enhancing ability of the reported technique on different imaging modalities.

We also noticed that the enhancement of our technique on experiment data (Fig. 4c) is not as superior as that on simulated data (Fig. 4b). We consider the reason may arise from the fact that in practical FPM experiments, besides the noise coming from the detection part, there also exist noise and aberrations from the illumination part and optical path. Especially for the FPM imaging modality that requires multiple acquisitions, there may be more aberrations such as non-uniform light flux and LED misalignment[39]. In this regard, it is our future work to investigate deep into these factors. One possible solution is to incorporate the reported enhancement technique into the iterations of FPM optimization, which may help reduce error accumulations and improve robustness.

### Experiment on high-speed scenarios

SPAD imaging holds unique advantages in capturing ultra-fast scenes under limited illumination conditions. Our SPAD arrays can achieve a minimum integration time of 20 ns to capture 1-bit images. We demonstrate the effectiveness of SPAD imaging for high-speed, microsecond timing scenarios in Fig. 5.

The first demonstration involves recording a rapidly rotating fan. In this experiment, we captured images of the spinning fan using an experimental setup shown in Fig. 5a. The setup consists of an objective lens, an LED light source (GCI-060401, 3W electrical power), and the SPAD arrays. The integration frame time was set to 1 microsecond. We observed that there was noticeable noise and aberration in the raw single-photon images due to heavy SPAD

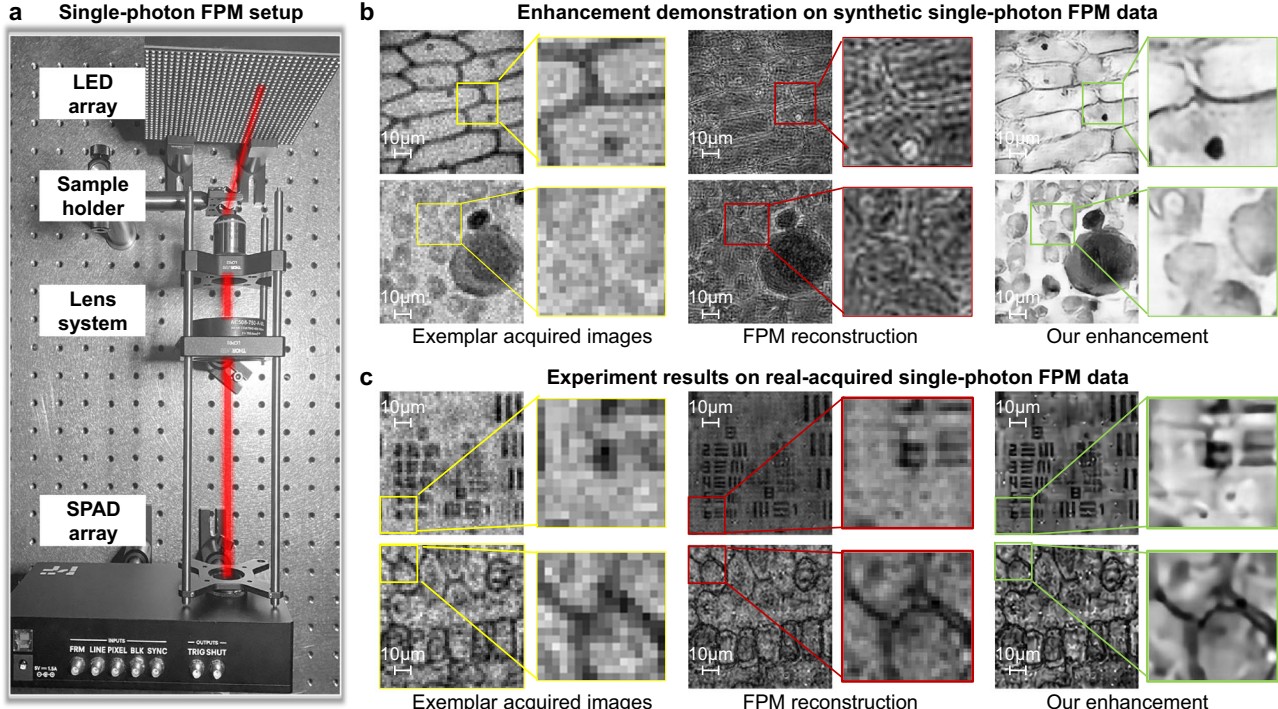

**Fig. 4 | Fourier ptychographic microscopy (FPM) experiment using single-photon detection. a** The proof-of-concept FPM setup using a SPAD array camera with a Light Emitting Diode (LED) array. **b**, **c** presents the enhancement results using synthetic data and real-acquired FPM data, respectively. The left column shows the images (32 × 32 pixels) acquired under vertical illumination (7 bits under 3 μs exposure time), and the middle column presents the FPM reconstructed images (each using 81 acquired images to recover 192 × 192 pixels). The right column shows further-enhanced results (384 × 384 pixels) using the reported technique, with the FPM reconstruction as inputs.

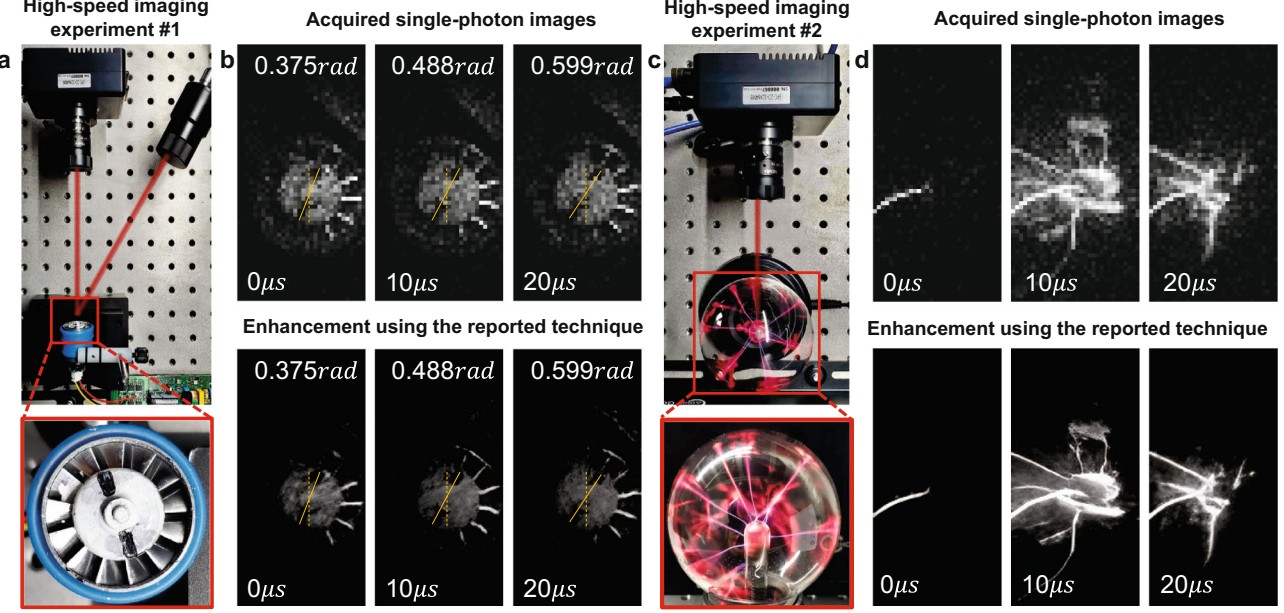

**Fig. 5 | Single-photon imaging experiment of high-speed scenarios. a** The single-photon imaging setup used to record the rotating fan. **b** The captured single-photon images at microsecond time intervals, along with their corresponding enhanced results. **c** The single-photon imaging setup used to capture the arc. **d** The obtained single-photon images of the arc at microsecond time intervals, and their corresponding enhancement results.

noise, making it difficult to distinguish the rotating blade angle of the fast fan. We applied the reported technique to enhance the recorded SPAD images. The results showed improvement compared to the initial recordings. The background appeared smoother, and the super-resolution details were further enhanced. We have measured the rotational speed to be 0.0107 rad/μs, which is approximately equivalent to 102,000 RPM (revolutions per minute). This measured speed is consistent with the manufacturer's instructions, which state a rotating speed of about 100,000 revolutions per minute.

The second demonstration features an electrostatic ball, as presented in Fig. 5c. When activated, the circuit generates a high-frequency electric field that illuminates the thin gas inside the sphere. A high voltage alternating current is applied to the sphere via a central electrode. This energy ionizes the gases, producing positively charged ions and plasma. The high voltage from the electrode then creates an arc through the plasma to the edge of the sphere, causing it to glow. We conducted recordings of the glowing process at microsecond intervals to enhance the understanding of the high-speed arc glowing process. Through observations, we noted that the duration of the arc glowing period is approximately $4\mu s$, which is consistent with the statement in ref. 41. We provided a high-speed single-photon imaging demonstration video in Supplementary Movie 1.

## Discussion

In this work, we introduced deep learning into single-photon detection, and presented a enhancing technique for large-scale single-photon imaging. By tackling the big challenge of single-photon enhancement with extremely low resolution, low bit depth, complex heavy noise and lack of image datasets, the reported technique realizes simultaneous single-photon denoising and super-resolution enhancement (up to 4 × 4 times). A series of experiments on both macro and micro setups validate the reported technique's state-of-the-art large-scale single-photon imaging performance. Besides, we also built two application setups of microfluidic inspection and Fourier ptychographic microscopy to further validate the strong adaptation ability of the reported technique for various imaging modalities.

The state-of-the-art large-scale single-photon imaging performance of the reported technique builds on several innovations, including the physical multi-source noise modeling of SPAD arrays, the construction of two single-photon image datasets (one for noise calibration and another for network training), and the design of a deep transformer neural network with a content-adaptive self-attention mechanism for single-photon enhancement. The reported noise model studied the entire photon flow process of single-photon detection, and considered multiple noise sources that degrade single-photon imaging quality. The noise sources include shot noise from photon incidence, fixed-pattern noise from SPAD array's photon response, dark count rate, afterpulsing and crosstalk noise from blind electron avalanche, and deadtime noise from the quenching circuit. The first single-photon image dataset of 3600 single-photon images was collected with different illumination flux and bit depth, to calibrate the statistical parameters of the multi-source physical noise model. Employing the calibrated noise model, the second single-photon image dataset was synthesized, containing 1.1 million images over 17250 scenes, providing image pairs of low-resolution single-photon ideas and corresponding high-resolution ground truth ranging from thousand to mega pixels. This strategy saves great efforts to collect and register paired data, and makes it possible to study latent signal features of single-photon data. The dataset was applied to train the transformer network for single-photon image enhancement, providing a smooth background and fine details benefiting from its self-attention and gated fusion mechanism.

In the workflow, the multi-source noise model parameters were calibrated employing a set of collected images acquired using a specific SPAD camera. Although the reported noise model is generalized and applicable to various single-photon detection schemes, the noise parameters of different SPAD arrays may deviate from each other, even for the same version of cameras. In this regard, the automatic calibration of different SPAD arrays is worthy of further study. Besides, we consider that the transfer learning technique can be adapted to other single-photon detection hardware and settings[42]. This would save laboursome data collection and parameter calibration efforts for different experiments.

Furthermore, big model training has gotten a big development for computer vision tasks[43]. It may be an effective method to generate synthetic datasets based on the reported physics model to train a big model for SPAD imaging.

Besides direct enhanced imaging, single-photon detection has also been applied in multiple computational sensing modalities for broader applications, such as non-line-of-sight imaging[44,45] and quantum key distribution[46]. In such schemes, we can integrate the reported enhancing technique with the sensing framework to achieve global optimum[47,48]. The technique may work as an enhancing solver in the alternating optimization process to improve resolution and attenuate noise and aberrations[47], preventing error accumulation for global optimization.

The photon detection efficiency of the SPAD array is varied at different wavelengths. This property provides two hints for us to further develop the reported technique. First, we can further consider this wavelength-dependent noise in our multi-source physical noise model, which can compensate for the degradation introduced by varied photon efficiency. Second, we can employ various photon efficiency to retrieve spectral information of incident light, opening research avenues on single-photon multispectral imaging that would benefit a set of single-photon applications[49].

Besides single-photon detection sensitivity, another highlighted ability of SPAD is its picosecond-scale time gating, indicating the time stamp of photon arrivals. In our present work, we did not consider time gating in different applications such as 3D LiDAR imaging capabilities for autonomous vehicles[23] and fluorescence lifetime imaging in microscopy. Prior to the availability of the reported method, addressing issues with commercially available 64 × 32 resolution SPAD for LiDAR systems was difficult, as they struggled to perform accurate and fast detection due to limited resolution. Using the reported approach, we can assist LiDAR systems in obtaining higher resolution and more precise scene information for enhanced detection. Furthermore, data collection in complex environments, such as rainy, foggy, or hazardous conditions, poses even greater challenges. Tackling these scenarios is vital for safety-related tasks. In this context, the reported technique presents an alternative solution for LiDAR systems confronted with these situations. In our future work, it is worth studying time gating in our enhancement framework to improve depth-selective resolution. Besides, considering time stamp into the multi-source physical noise model may further help improve noise robustness and enhancing ability.

## Methods

### Noise modeling of SPAD arrays

As shown in Fig. 1a, the noise sources of SPAD arrays include signal-dependent shot noise from photon incidence, fixed-pattern noise from SPAD array's photon detection efficiency (PDE), dark count rate, afterpulsing and crosstalk noise from electron avalanche, and deadtime noise from circuit quenching.

In the incident process that photons enter the photosensitive surface of SPAD arrays, photon arrival is a stochastic process due to the quantum properties of light, known as the shot noise $N_{shot}$. During a certain integration time of the detector, the incident photon number $k$ subjects to the Poisson distribution as $p(k) = \frac{e^{-\chi}\chi^k}{k!}$, where $\chi$ is the expectation of incident photons in a unit time (namely the latent light signal). Considering the unique avalanche circuit of SPAD arrays that detects at most one photon during the integration time, we apply the Poisson distribution with probability $p(\text{Poisson}) = 1 - p(k = 0)$, where $p(k) = \frac{e^{-\chi}\chi^k}{k!}$. In this regard, the shot noise is independent of specific detectors, and its parameters do not need to be calibrated. We considered shot noise in our synthesized large-scale single-photon image dataset for network training.

SPAD arrays absorb incident photons and generate electric charges through photoelectric conversion. In this process, each SPAD pixel

responds with a certain probability, and an avalanche occurs under a certain probability to form a saturation current for a new photon count. The probability that the above process occurs is termed photon detection efficiency (PDE), which is defined as the ratio between the number of incoming photons and the number of output current pulses. Since PDE is mainly related to manufacture craft and hardware settings, it is approximated to be a fixed probability distribution, meaning that the fixed-pattern noise ($N_{fp}$) probability of each pixel is assumed to be a constant once it is calibrated. In our workflow, we employed the PDE data provided by manufacturer. In the photoelectric conversion process, SPAD arrays would also excite electrons due to the thermal effect even in darkness. These electrons are amplified by the avalanche circuit to form a saturation current, resulting in dark count rate noise. The dark count rate noise $N_{dcr}$ subjects to the Poisson distribution[50], with the expectation to be calibrated.

Followed the photoelectric conversion, the electrons are amplified by an avalanche. In the electron avalanche amplification process, the carriers pass through the P-N junction and may be trapped by the conductor. In such a case, the saturation current is relayed, and the detector will count an additional detection event with a certain probability, denoted as afterpulsing noise $N_{ap}$. The existing studies of afterpulsing noise[51,52] consider its probability to be a power function or exponential decay with time. Since we only consider its statistical effect related to the manufacture of each pixel unit, we simplify the model to be a fixed probability distribution map, and assume that its effect of the previous frame may only apply to the current frame. Besides the electrical interference, the avalanche current in one pixel may trigger the surrounding pixels due to the photons emitted by hot carriers. This process produces crosstalk noise $N_{ct}$, which is assumed to follow a fixed probability distribution similar to afterpulsing noise[53].

To prevent self-sustaining current from damaging the circuit, SPAD arrays implement a quenching operation that adjusts the bias of P-N junction to the breakdown voltage after the avalanche during each integration time. In this process, each pixel does not respond to additional incident photons. As a result, this process brings deadtime noise to photon count and keeps it being zero. In our experiments, we set the integration time to be 20 ns, the dead time to be 60 ns, and the readout time to be 80 ns. In this way, we can eliminate the negative influence of asynchronous deadtime noise of each pixel. Therefore, we do not include deadtime noise in the following modeling and calibration phases.

To sum up, the multi-source physical noise model of SPAD arrays is described as

$$N = N_{shot} + N_{fp} + N_{dcr} + N_{ap} + N_{ct} + N_{dt} \tag{1}$$

As stated above, the negative influence of $N_{shot}$, $N_{fp}$ and $N_{dt}$ is tackled by either data-driven processing, employing manufacture parameters or adjusting hardware settings. The model parameters that are required to calibrate include the expectation of $N_{dcr}$, the fixed probability $p_{ap}$ of $N_{ap}$, and the expectation of $N_{ct}$.

## Noise parameter calibration

We acquired 60000 single-photon images (1 bit) of dark field without illumination. In such a case, we considered shot noise $N_{shot} \approx 0$ and fixed-pattern noise $N_{fp} \approx 0$. Besides, the deadtime noise is also assumed $N_{dt} \approx 0$, because it mainly describes the response to incident light in the quenching process. In this regard, the noise formation model of the dark-field images can be summarized as

$$I_{dark}(x,y,n) = N_{dcr}(x,y,n) + p_{ap}(x,y)I_{dark}(x,y,n-1) \\ + p_{ct}(x,y)U(I_{dark}(x,y,n)), \tag{2}$$

where $I_{dark}(x,y,n)$ denotes the detected signal at the pixel location $(x,y)$ in the $n$-th frame, $N_{dcr}(x,y,n)$ is the dark-count rate noise map that follows Poisson distribution, $p_{ap}$ is the fixed probability map of afterpulsing noise, $p_{ct}$ represents the Poisson probability of crosstalk noise, and $U(I_{dark}(x,y,n))$ denotes neighboring detected signal of pixel $(x,y)$.

To calibrate the above multiple noise parameters, we first extract the noise maps of afterpulsing noise $p_{ap}(x,y)$ and crosstalk noise $p_{ct}(x,y)$ according to their generation mechanism. Specifically, because the detected photon number of dark images is small, two consecutive detection events at the same pixel location is most likely afterpulsing noise. Under this assumption, we obtained the intensity of afterpulsing at each pixel $(x,y)$ in the $n$-th frame as $I_{ap}(x,y,n)$, provided that the signals of the previous frame and the current frame at the same pixel are the same as 1. Similarly, we can extract the map of crosstalk noise $I_{ct}(x,y,n)$, considering that the neighboring pixels in the same frame are both 1 if a crosstalk event occurs.

As stated in the technical manual provided by the SPAD manufacture, the typical probability of afterpulsing events is 1–2 orders of magnitude higher than that of crosstalk events. Therefore, we prioritize afterpulsing events during the calibration process. When both afterpulsing and crosstalk events are observed at a single pixel, we classify it as an afterpulsing event rather than a crosstalk event. By doing so, we can effectively ignore the influence of crosstalk events while ensuring that afterpulsing events receive the necessary attention. This strategy helps alleviate the problem of redundant calibration and enables us to obtain accurate parameter values.

Using the sub-noise maps, the fixed probability of afterpulsing noise is calculated as

$$p_{ap}(x,y) = \frac{\sum_n I_{ap}(x,y,n)}{\sum_n \left( I_{dark}(x,y,n) - I_{ap}(x,y,n) - I_{ct}(x,y,n) \right)}, \tag{3}$$

and the crosstalk noise probability is

$$p_{ct}(x,y) = \frac{\sum_n I_{ct}(x,y,n)}{\sum_n \left( I_{dark}(x,y,n) - I_{ap}(x,y,n) - I_{ct}(x,y,n) \right)}. \tag{4}$$

Followed the calibration of afterpulsing noise and crosstalk noise, the dark count rate noise is approximated as

$$N_{dcr}(x,y) = \frac{\sum_n \left( I_{dark}(x,y,n) - I_{ap}(x,y,n) - I_{ct}(x,y,n) \right)}{n_{total}}, \tag{5}$$

where $n_{total}$ is the number of acquired frames for calibration. In our implementation, we set $n_{total} = 60000$. More details of generating an synthetic single-photon image dataset based on the calibrated noise model is presented in Supplementary Note 7. We further provide our SPAD arrays' sampling/ISP scheme and simulation strategy in Supplementary Note 8.

## Network structure

We designed a gated-fusion transformer network for single-photon enhancement. The network is inspired by the Swin-Transformer structure[30], whose shift-window mechanism can effectively reduce network parameters by more than one order of magnitude compared to the conventional transformer networks[34]. On this basis, our network further introduces dense connections among different Swin-Transformer layers (STL), enabling long-distance dependency modeling and full-frequency information retrieval. As shown in Fig. 5a, the network consists of three modules, including the shallow feature extraction module, the deep feature fusion module, and the image reconstruction module. Compared with the conventional convolutional networks, the gated fusion transformer network maintains the

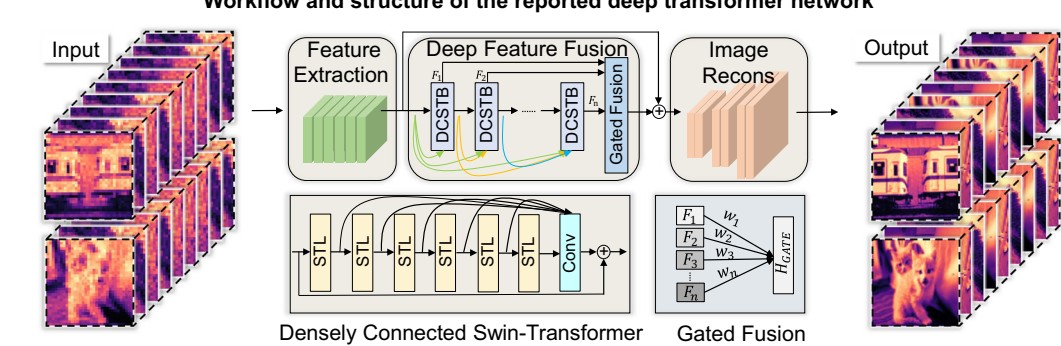

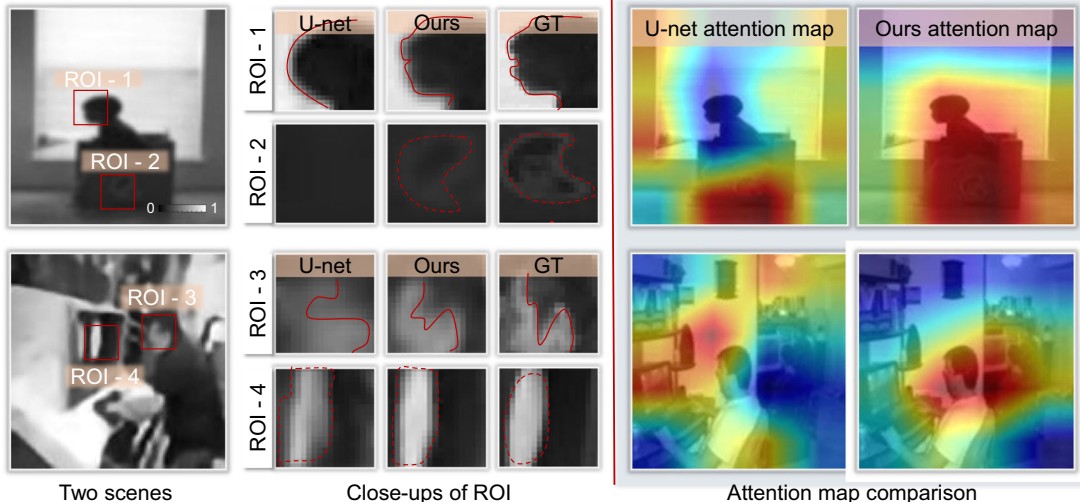

**Fig. 6 | Illustration of the reported deep transformer network for high-fidelity large-scale single-photon imaging. a** The workflow and structure of the reported network. The STL refers to Swin-Transformer layers, and the DCSTB refers to densely connected Swin-Transformer blocks. **b** The enhancement comparison between CNN-based U-net network and the reported transformer-based network. Benefiting from the transformer structure, our network realizes spatial-variable convolution that helps pay more attention to the regions of interests (ROI) and fine details, as the attention maps on the right side validate.

following advantages: 1) The content-based interactions between image content and attention weights can be interpreted as spatially varying convolution, and the shift window mechanism can perform long-distance dependency modeling. 2) The gated fusion mechanism can deeply dig medium-frequency and high-frequency information at different levels, enabling the prevention of losing long-term memory as the network deepens and enhancement of fine local details.

## Shallow feature extraction

Given a low-quality image $I_{LQ} \in \mathbb{R}^{h*w*c_{in}}$ ($h$, $w$ and $c_{in}$ are the image's height, width and channel number), we use the shallow feature extractor $H_{FE}(*)$ to explore its low-frequency features $F_0 \in \mathbb{R}^{h*w*c}$ as

$$F_0 = H_{FE}(I_{LQ}). \qquad (6)$$

This module is composed of convolution, batch normalization and activation layers (specific parameters and settings are presented in Supplementary Note 3). The convolution layers are applied for pre-liminary visual processing, providing a simple way to map the input image space to a higher-dimensional feature space. Besides the fol-lowing deep fusion operation, the output of this module is also linked to the final image reconstruction module, so that the target's low-frequency information can be well preserved in final reconstruction.

## Deep feature fusion

Next, we use several densely connected Swin-Transformer blocks (DCSTB) to extract different levels of medium-frequency and high-frequency features $F_i \in \mathbb{R}^{h*w*c}(i=1,2,\ldots,n)$ from $F_0$, denoted as

$$F_i = H_{DCSTB}(F_0, F_1, \ldots, F_{i-1}), \qquad (7)$$

where $H_{DCSTB}$ represents the $i_{th}$ DCSTB operation. Compared to the conventional convolution blocks such as U-net, the transformer-structure blocks realize spatial-variable convolution that helps pay more attention to the regions of fine details and interests, as validated in Fig. 5b. Consequently, such blocks help recover more high-frequency information that is beneficial to enhancing imaging resolution.

The last layer of the deep feature fusion module is the Gated Fusion layer, which fuses the outputs of different DCSTB opera-tions with adaptively different weights. The process can be described as

$$F_{DF} = H_{GATE}(F_1, F_2, \ldots, F_n) = w_1F_1 + w_2F_2 + \ldots + w_nF_n, \qquad (8)$$

where $F_{DF}$ represents the multi-level deep fusion features output by the gated fusion layer, and $w_n$ represents the weight parameters during gated fusion for different levels of feature, which are adaptively adjusted through backpropagation during network training. Such a module structure is conducive to deep mining of different levels of

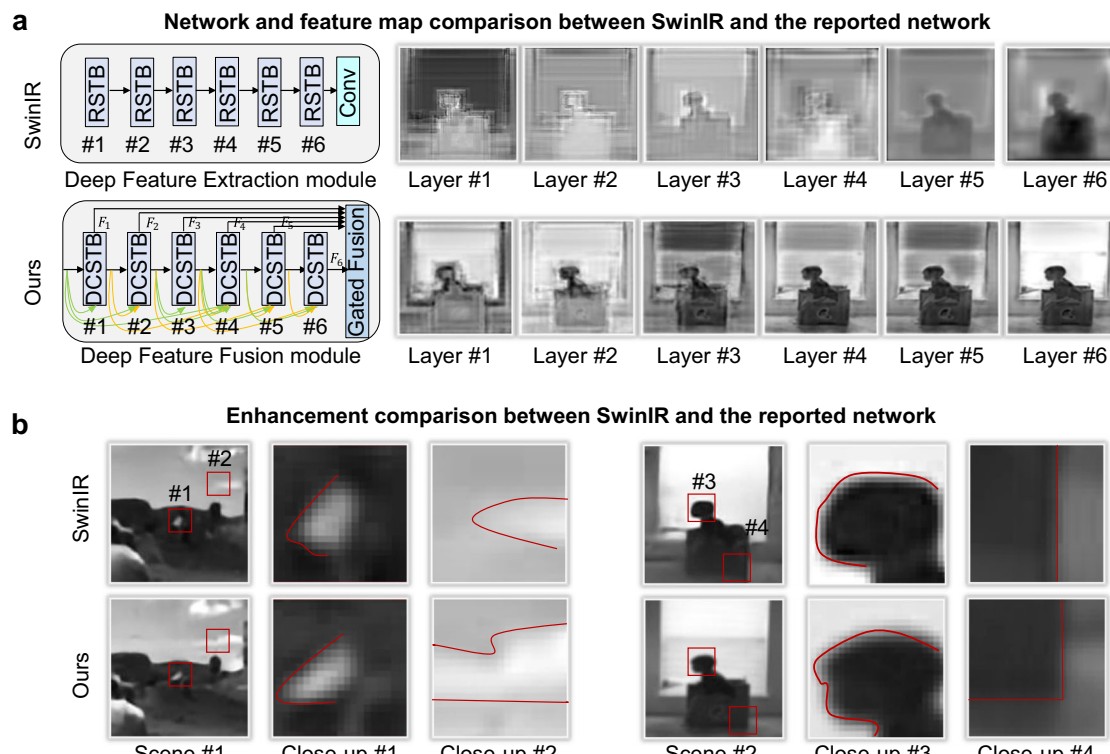

**Fig. 7 | The enhancement comparison between the SwinIR network and the reported network. a** The comparison of key feature modules and feature maps. The RSTB refers to residual swin-transformer block. We introduce dense connection and gated fusion among different feature blocks, which helps preserve different levels of medium-frequency and high-frequency information, and keeps long-term memory as network deepens. The intermediate feature maps shown on the right side clearly validate such superiority. **b** The enhancement comparison of fine details. Benefiting from the deep feature fusion mechanism, the reported network is able to recover more details.

medium-frequency and high-frequency information, which prevents losing long-term memory as the network deepens and enhances local details[54], as validated in Fig. 6b.

## Image reconstruction

We retrieve high-quality single-photon images by aggregating shallow features and multi-level deep fusion features. The operation is described as

$$I_R = H_{REC}(F_0 + F_{DF}). \tag{9}$$

The shallow features $F_0$ are mainly low-frequency image features, while the multi-level deep fusion features $F_{DF}$ focus on recovering lost medium-frequency and high-frequency features. Benefiting from the long-term skip connections, the Gated Fusion Transformer network can effectively transmit different-frequency information to final high-quality reconstruction. Different from the SwinIR network[55] that lacks the densely connected gated fusion structure (as shown in Fig. 7a), the reported network helps preserve and compensate the key medium and high-frequency feature signals, and enriches the image's local details (as the intermediate feature maps validate in Fig. 7a). In addition, sub-pixel convolution is applied in the reconstruction block to further upsample the feature map for single-photon super resolution. More network structure details are presented in Supplementary Note 10.

## Loss function

We designed a hybrid loss function consisting of $L_1 - norm$ loss, perceptual loss and SSIM loss, to train the Gated Fusion Transformer network. The $L_1 - norm$ loss calculates the absolute distance between two images as $Loss_{L_1}(I_R, I_G) = ||I_R - I_G||_{L_1}$, where $I_R$ represents the reconstructed image by the network, and $I_G$ denotes its ground truth.

The perceptual loss is defined as the $L_2 - norm$ distance between feature maps output by the pool-3 layer of a VGG19 network pretrained on ImageNet as $Loss_{PER}(I_R, I_G) = ||\varphi(I_R) - \varphi(I_G)||_{L_2}$, where the $\varphi(\cdot)$ operation extracts feature maps. The perceptual loss regulates different-frequency similarity in the feature space. The SSIM loss is calculated as $Loss_{SSIM} = 1 - SSIM(I_R, I_G)$, which further regulates the two images' similarity in the structural domain. To sum up, the loss function for network training is

$$Loss(I_{RHQ}, I_{HQ}) = \alpha Loss_{L_1}(I_{RHQ}, I_{HQ})$$
$$+ \beta Loss_{PER}(I_{RHQ}, I_{HQ}) + \gamma Loss_{SSIM}(I_{RHQ}, I_{HQ}). \tag{10}$$

where $\alpha$, $\beta$ and $\gamma$ are hyperparameters balancing the three loss parts. In our implementation, these hyperparameters were set as $\alpha = 0.1, \beta = 10$ and $\gamma = 100$ after careful network tuning. Furthermore, we have presented the results of our transfer learning approach for further improvement in Supplementary Note 9.

## Network training

We randomly selected 505 image pairs from the synthetic large-scale single-photon image dataset as the testing set, and the remaining 16620 image pairs were used for network training. All the images were cropped to $32 \times 32$ pixels when input into the network. We implemented the network on Ubuntu20 operating system using the Pytorch framework, and trained 1000 epochs until convergence using Adam optimization on NVIDIA RTX3090 with the Batch size set to 24. We set the initial learning rate as 0.0003, which was decreased by 10% for every 100 epochs. The default weight decay was set to 0.00005, and the Adam parameters of $\beta 1$ and $\beta 2$ were set to 0.5 and 0.999, respectively. We used RTX3090 GPU for network implementation on the Python

and Pytorch frameworks. The average time to enhance a single SPAD image from 32*64 to 128*256 is 0.04 s. Other details of network training are presented in Supplementary Note 4.

The minimum data generated in this study have been deposited in the Figshare database under accession code https://doi.org/10.6084/m9.figshare.23966922.v1. The complete data are available under restricted access for the funded project requirements, access can be obtained by the corresponding author within one month of the reasonable request to the corresponding author.

## Data availability
The minimum data generated in this study have been deposited in the Figshare database under accession code https://doi.org/10.6084/m9.figshare.23966922.v1. The complete data are available under restricted access for the funded project requirements, access can be obtained by the corresponding author within one month of the reasonable request to the corresponding author.

## Code availability
The demo code of the reported technique is available from https://github.com/bianlab/single-photon.

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

## Acknowledgements

This work was supported by the National Natural Science Foundation of China (Nos. 61971045, 61827901, 61991451).

## Author contributions

L.B. conceived and designed the project. H.S., X.Y., R.H., and L.Y. designed and implemented the experiments. L.P. and H.S. developed the network and conducted image enhancement. L.B., H.S., and L.P. wrote the manuscript with input from all authors. X.C., C.Z., T.Q., D.Z., and J.Z. helped revise the manuscript. J.Z. and L.B. supervised the work.

## Competing interests

L.B., H.S., and J.Z. hold patents on technologies related to the techniques developed in this work (China patent numbers ZL 2020 1 1157082.2) and submitted related patent applications. The remaining authors declare no competing interests.
