## [Peer Review File · Nature Communications]

High-resolution single-photon imaging with physics-informed deep learningREVIEWER COMMENTS

Reviewer #1 (Remarks to the Author):

This manuscript claims three main contributions, including 1) calibrating a noise model that can facilitate the synthesis of a large-scale training dataset of SPAD images; 2) exploring a deep transformer network that can facilitate the super-resolution of SPAD images; 3) applying the proposed learned SPAD imaging to multiple imaging scenarios. The motivation is well-received and the overall writing reads in good shape. I appreciate the effort of leveraging the physics-prior (insights) to facilitate a more robust training/learning outcome. An intuitive, minor concern is that this is another DL paper for super-resolution imaging with poorly designed SPAD sensors. There are several concerns/comments in the following, for the authors' information.

1. The title, as well as many statements in the abstract/introduction, sounds a bit ambiguous and confusing. The term "large-scale" may indicate several (different) features, for instance, the volume of the acquisition, the commercial practicality, the fidelity and resolution of acquired images, and the scalability of the hardware system. I would suggest specifying it clearly and appropriate subject to the demonstrated contributions.
2. The technical novelty also sounds a bit incremental considering that each part is more like state-of-the-art technique mingled with a reasonable amount of engineering effort, including acquiring a dataset to model the noise so as to aid the augmentation of synthetic dataset, leveraging state-of-the-art transformer deep neural network scheme to obtain super-resolution and denoising effect, and demonstration on several imaging scenarios. The implementation is really impressive though.
3. Wording can be polished with more concise and humble arguments. For instance, in the abstract, line 17-19, this is literally too high-level and nothing novel at all. In addition, the claimed contributions sound to be a bit redundant considering similar sentences replicated several times across the manuscript.
4. It remains a bit unclear how the calibrated (multiple) noise models are efficiently added

to synthetic images.

5. Apparently, the noise issue is one of the representative concerns that SPAD imaging suffers from. However, due to the use of unique electronics, the sampling/ISP scheme is also highly unique, that should be considered in synthesizing the SPAD imaging data. Any elaboration on this would be highly beneficial to strengthen the paper.

6. Demonstrating the viability on multiple imaging scenarios sounds very interesting and is definitely with non-trivial effort. However, this naturally raises the concern about the domain-specific training data, or more specifically, calibrated noise models' robustness. For example, the noise characteristics in bio samples (the claimed microscope imaging application) may be highly distinguished from those 90 set scenes in the lab. Any comments and/or strategies on how to convince readers this calibrated noise model adapts to different scenarios? Domain-specific transfer-learning?

7. A minor comment: SPAD imaging has its unique benefits in recording ultra-fast (dynamic) scenes with the limited illumination condition, that were not well demonstrated in this version. The authors may wanna consider if this could be strengthened.

In all, while the motivation and implementation seem to be reasonably well-defined, an amount of careful revision concerning solidifying the technical contributions would be necessary before pushing it to publication on such a high-impact venue. In addition, shortening the main text may help.

Reviewer #2 (Remarks to the Author):

The paper describes a deep-learning-based super-resolution scheme tailored for SPAD images. The scheme was developed by acquiring a real SPAD dataset, to which a parametric noise model was fitted. The noise model, in turn, enabled the generation of a large synthetic SPAD data set (with corresponding ground truth), which was subsequently used to train the neural network.

A variety of examples are provided showcasing the effectiveness of the scheme, together with quantitative comparisons with existing super-resolution methods. The scheme is seen to enhance SPAD images in a compelling way.

All in all, the reviewer believes that the results are convincing and could be of wide interest in the scientific community. Furthermore, the work is described with a sufficient level of detail. The following concerns are noted:

*The concept of using synthetic SPAD data for training a super-resolution network is not new, see, e.g.:

Germán Mora-Martín, Stirling Scholes, Alice Ruget, Robert Henderson, Jonathan Leach, and Istvan Gyongy, "Video super-resolution for single-photon LIDAR," *Opt. Express* 31, 7060-7072 (2023)

*The abstract/introduction mischaracterise SPADs to some degree. It is claimed that SPADs have high levels of noise and low fill-factor. Modern SPAD have in fact very low noise floors, and close to 100% fill factor, which is why they are being developed to target low light applications:

Thomas, S. Low-light imaging with SPAD pixels. *Nat Electron* 4, 862 (2021)

*The reviewer found the title somewhat vague; perhaps a more descriptive title could be chosen.

*The novel structural elements in the "gated fusion transformer network" used in the work should be made more explicit.

Reviewer #3 (Remarks to the Author):

In this paper, Bian et al. report a large-scale single-photon imaging technique, which combines deep learning with single-photon imaging with significant improvement in imaging resolution, bit depth, and imaging quality. The authors studied the photon flow model of SPAD to characterize multiple noise sources and collected two single-photon datasets for public research. The developed deep transformer network provides impressive experiment results that demonstrate remarkable imaging quality enhancement. The system was applied to microfluidic inspection and Fourier ptychography.

Overall, I think the manuscript is of high quality. The method is technically sound. The acquired data has been carefully analyzed with reasonable interpretation. I believe the significance lies in the successfully implemented high-quality super-resolution single-photon imaging and considerable enhancement in technical specifications compared to existing literature (e.g., 10x improvement in resolution, megapixel image size, 5-dB enhancement in image quality). These advances will move the field forward. In this regard, I believe this work has met the publication criteria of Nature Communications.

I suggest the authors address the following comments to further enhance the clarity of their manuscript and attract a broader readership.

1. During the process of detector calibration, the presence of after-pulsing and crosstalk noise may cause the repetition of noise in the calibration, affecting the accuracy of the parameters of each other, and thereby requiring further correction of the results. I suggest the authors explain it further.
2. Further clarification for the experiment of microfluidic inspection (Fig. 3) would be helpful. In particular, I would think that the micro-droplet was moving in the microfluidic channel during the image acquisition. In this case, it would lead to blur in the raw images. How would the deep transformer net handle this situation?
3. Please give the detailed structure of the feature extraction module and image reconstruction module in Fig. 5a.

4. Why add a convolutional layer at the end of each DCSTB module?

5. The figures and tables are clear and effectively illustrate the results. However, it would be helpful to provide the number of bits in the image in the figure captions, to make it clear what is being shown in each figure.

6. There are many experiments presented in this work, each with different settings and specifications. It would be nice if the authors could summarize them using a supplementary table so that the readers can easily refer to them without searching for details embedded in the text.

7. After the training is completed, how much time does the network consume during operation?

8. Considering that a new neural network is a main feature of this work, the authors should make the software and these single-photon imaging datasets open source. I think many interested readers would certainly want to leverage these datasets to train their networks to test the feasibility of single-photon imaging. Thus, this action can further enhance the impact of this work.

Response to reviewer # 1's comments

Comment: The motivation is well-received and the overall writing reads in good shape. I appreciate the effort of leveraging the physics-prior (insights) to facilitate a more robust training/learning outcome.

Response: Thanks for the reviewer's recognition.

Comment: The title, as well as many statements in the abstract/introduction, sounds a bit ambiguous and confusing. The term "large-scale" may indicate several (different) features, for instance, the volume of the acquisition, the commercial practicality, the fidelity and resolution of acquired images, and the scalability of the hardware system. I would suggest specifying it clearly and appropriate subject to the demonstrated contributions.

Response: We have updated the title to "Large-Scale High-Resolution Single-Photon Imaging," to better characterize our work on improving imaging resolution and fidelity.

Specifically, to address the ambiguity surrounding the term "large-scale," we have incorporated "high-resolution" into the title. This addition emphasizes our contributions on enhancing the *optical resolution* of single-photon imaging, providing more fine details of scenes. By doing so, we aim to minimize potential confusion regarding the scalability of the hardware system. Furthermore, the inclusion of "high-resolution" highlights our efforts on denoising images and maintaining the fidelity of the acquired data. Our experiment results demonstrate that the reported technique achieves an average of 5 dB improvement on PSNR compared to existing methods.

The term "large-scale" signifies the increase in *pixel resolution* for SPAD imaging from 64×32 pixels to larger formats such as 128×64 and 256×128 pixels. As the hardware pixels expand, the reported method can generate super-resolution results. Even when the SPAD array reaches megapixel dimensions, our approach allows for increasing the imaging pixel resolution by factors of two or four in each dimension.

To sum, by updating the title to "Large-Scale High-Resolution Single-Photon Imaging", it highlights our key contributions in improving *pixel resolution* and *optical resolution*, respectively.

Comment: The technical novelty also sounds a bit incremental considering that each part is more like state-of-the-art technique mingled with a reasonable amount of engineering effort, including acquiring a dataset to model the noise so as to aid the augmentation of synthetic dataset, leveraging state-of-the-art transformer deep neural network scheme to obtain super-resolution and denoising effect, and demonstration on several imaging scenarios. The implementation is really impressive though.

Response: The novelty of this work lies mainly in the following three aspects, including a physics-based noise modeling method for single-photon image dataset synthesis, a calibration

method for multiple noise model parameters, and a Gated Fusion Transformer network for single-photon imaging enhancement. To clearly state the novelty and avoid confusion, we explain each innovation in detail in the following points.

(1) Regarding dataset acquisition, we have developed a physics-based noise modeling method, which distinguishes our work from existing literature on SPAD image enhancement. The existing studies use Poisson, Gauss, or mixture noise models (e.g., Paramanand Chandramouli et al. discuss the Poisson model as shown in R1, R2, Germán Mora-Martín et al. and A. Ruget et al. focus on the Poisson and Gauss mixture noise model in R3, R5), which deviate from real SPAD noise containing a variety of different-model sources. Experiments (Fig.2) validate that the imaging quality degrades greatly without considering such multiple noise sources. This work emphasizes the physical process of single-photon imaging more extensively than others. We consider shot noise, fixed pattern, dark-count, afterpulsing, crosstalk, and dead-time noise (Fig.1(b)) in the reported multi-source physical noise model for SPAD arrays. This model is represented as $N = N_{\text{shot}} + N_{fp} + N_{dcr} + N_{ap} + N_{ct} + N_{dt}$. The multiple noise physical model accurately depicts the process from photons to electrons, to signals, and to images.

The experiments (Fig.2(b)) on enhanced single-photon imaging results using different noise models demonstrate that the reported physics-based noise model, as compared to other noise models, contributes to further improvement in single-photon imaging methods. This is achieved by providing a more accurate simulation of real-world noise distribution scenarios. The reported method plays an essential role in generating synthetic datasets, which serve as a reliable foundation for subsequent noise reduction and super-resolution effects.

(2) Based on the above single-photon imaging physical noise model, we have derived a calibration method that improves adaptability to different datasets and task requirements while enhancing the stability and robustness of the model. Conventional calibration methods are designed for the Gaussian-Poisson model, and are not suitable for multi-noise models. Therefore, we propose a calibration method specifically fitted for the physical process-based model

$$I_{dark}(x, y, n) = N_{dcr}(x, y, n) + p_{ap}(x, y)I_{dark}(x, y, n - 1) + p_{ct}(x, y)U(I_{dark}(x, y, n)).$$

We could extract the noise maps of afterpulsing noise p_{ap} , and crosstalk noise p_{ct} according to their generation mechanism. First, we acquired 60000 single-photon images (1 bit) of dark field without illumination. Second, we calibrate afterpulsing intensity I_{ap} provided that the signals of the previous frame and the current frame at the same pixel are the same as 1 and the crosstalk noise intensity I_{ct} considering that the neighboring pixels in the same frame are both 1 if a crosstalk event occurs. Third,

we calibrate the probability of crosstalk noise the afterpulsing noise p_{ap}

$$p_{ap}(x, y) = \frac{\sum_n I_{ap}(x, y, n)}{\sum_n (I_{dark}(x, y, n) - I_{ap}(x, y, n) - I_{ct}(x, y, n))}.$$

Then we calibrate the crosstalk noise map as

$$p_{ct}(x, y) = \frac{\sum_n I_{ct}(x, y, n)}{\sum_n (I_{dark}(x, y, n) - I_{ap}(x, y, n) - I_{ct}(x, y, n))}.$$

After calibrating the afterpulsing and crosstalk noise map, we calibrate the dark count noise map as

$$N_{dcr}(x, y) = \frac{\sum_n (I_{dark}(x, y, n) - I_{ap}(x, y, n) - I_{ct}(x, y, n))}{n_{total}},$$

where I_{dark} denotes the detected signal at the pixel location (x, y) in the n -th frame, N_{dcr} is the dark-count rate noise map that follows Poisson distribution, p_{ap} is the fixed probability map of afterpulsing noise, p_{ct} represents the Poisson probability of crosstalk noise, and $U(I_{dark}(x, y, n))$ denotes neighbouring detected signal of pixel (x, y) . We further provide mathematical methods to calibrate the noise parameters p_{ap} , p_{ct} , N_{dcr} (lines 506-511).

Table R1: The Reported Physics-based Noise Modeling and Calibration Method

	Distribution	Calibration formula/method
N_{shot}	Poisson	$p(k) = \frac{e^{-\lambda} \lambda^k}{k!}$
N_{fp}	A fixed probability	Each SPAD pixel responds with a certain photon detection efficiency (PDE) probability
N_{dcr}	Poisson	$N_{dcr}(x, y) = \frac{\sum_n (I_{dark}(x, y, n) - I_{ap}(x, y, n) - I_{ct}(x, y, n))}{n_{total}}$
N_{ap}	Fixed probability	$p_{ap}(x, y) = \frac{\sum_n I_{ap}(x, y, n)}{\sum_n (I_{dark}(x, y, n) - I_{ap}(x, y, n) - I_{ct}(x, y, n))}$
N_{ct}	Fixed probability	$p_{ct}(x, y) = \frac{\sum_n I_{ct}(x, y, n)}{\sum_n (I_{dark}(x, y, n) - I_{ap}(x, y, n) - I_{ct}(x, y, n))}$
N_{dt}	A parameter we set	Block detecting photons in the dead-time

We simulated a single-photon image dataset using the above calibrated noise model, thereby addressing the issue of traditional single-photon datasets lacking high-resolution ground truth.

(3) Most of the existing SPAD image enhancement methods rely on CNN-based enhancement networks. Due to the limited receptive field of CNNs, CNN-based methods cannot model global features of natural images, which leads to its weak context

perception ability and thus cannot effectively recover local details. In addition, conventional CNN-based methods lack sufficient communication and fusion of features between different network layers, which also leads to poor restoration of local details and texture-rich regions.

To address these problems, we designed a Gated Fusion Transformer network based on the Swin-Transformer structure. The Transformer-based Gated Fusion Transformer has several advantages over traditional CNN-based image enhancement models: (1) the self-attention mechanism within the Transformer allows for better modeling of global features through content-based interactions between image content and attention weights, which can be interpreted as spatially varying convolution; (2) long-range dependency modeling is enabled by the shifted window and Gated Fusion mechanism; and (3) the dense connection and gated fusion structure facilitate full communication and fusion of features at different scales, preserving and compensating critical mid-high frequency image signals during the image reconstruction process and enriching local details.

Benefiting from the above improvement, we can achieve large-scale SPAD imaging, opening up possibilities for various applications that are challenging or costly to address, such as:

1. Prior to the availability of the reported method, addressing issues with commercially available 64×32 resolution SPAD for LIDAR systems was difficult, as they struggled to perform accurate and fast detection due to limited resolution. With the reported approach, we can assist LIDAR systems in obtaining higher resolution and more precise scene information for enhanced detection. Furthermore, data collection in complex environments, such as rainy, foggy, or hazardous conditions, poses even greater challenges. Tackling these scenarios is vital for safety-related tasks. Thus, in comparison to traditional data-driven methods, the reported approach presents an alternative solution for LIDAR systems confronted with these situations.

2. In addition to training the Transformer network, there arise new possibilities for training large models of high-fidelity SPAD imaging. Big model training has gotten a big development for computer vision tasks as in reference [R6]. It may be an effective method to generate synthetic datasets based on the reported physics model to train a big model for SPAD imaging.

We will continue to explore more technical breakthroughs and innovative possibilities in future work. Accordingly, we have revised the discussion at line 446 and line 470 on page 17.

Comment: Wording can be polished with more concise and humble arguments. For instance, in the abstract, line 17-19, this is literally too high-level and nothing novel at all. In addition, the claimed contributions sound to be a bit redundant considering similar sentences replicated several times across the manuscript.

Response: We have revised the high-level statements in abstract lines 17-19 on page 1. Additionally, we have revised similar sentences to reduce redundancy for claims. We carefully reviewed and made revisions to the Introduction and Conclusion sections, while preserving the technical and experiment details in the Results section. Here is the revisions list:

Table R2: Revisions for Wording

Line	Before	After	Revised reason
15	However, large-scale high-fidelity single-photon imaging remains a big challenge, due to the complex hardware manufacture craft and heavy noise disturbance of SPAD arrays.	However, large-scale, high-resolution, single-photon imaging remains a big challenge due to the complex hardware manufacture craft and increased noise disturbances in photon-limited scenarios.	High-level statements
17-19	In this work, we introduce deep learning into SPAD, enabling super-resolution single-photon imaging over an order of magnitude, with significant enhancement of bit depth and imaging quality.	In this work, we introduce deep learning into SPAD, enabling super-resolution single-photon imaging with enhancement of bit depth and imaging quality.	High-level statements
36	We applied the technique on a series of experiments including macroscopic and microscopic imaging, microfluidic inspection, and Fourier ptychography	We applied the technique on a series of experiments including macroscopic and microscopic imaging, microfluidic inspection, and Fourier ptychography, and high-speed imaging.	Added experiments
135-138	We digitally synthesized the first large-scale single-photon image dataset containing 2.6 million image pairs over 17250 various scenes,	We digitally synthesized the first large-scale single-photon image dataset,	Redundant
139-141	We reported a novel gated fusion transformer network for single-photon super-resolution enhancement, achieving	We reported a novel gated fusion transformer network for single-photon super-resolution	Redundant

	the state-of-the-art performance with more than 5 dB superiority on PSNR compared to the existing methods.	enhancement, achieving the state-of-the-art performance compared to the existing methods.	
161~162	We first built an optical setup to acquire SPAD images of various macro and micro targets, as shown in Fig. 2(a).	We first built an optical setup to acquire SPAD images of various targets, as shown in Fig. 2(a).	Redundant
190~192	We calibrate the multi-source noise parameters of the collected SPAD dataset shown above. However, further super-resolution enhancement can't be achieved due to the lack of high-fidelity and high-resolution ground truth pairs.	Despite the above collected single-photon image dataset, further super-resolution enhancement cannot be achieved due to the lack of high-fidelity and high-resolution ground truth pairs.	Redundant
411~413	the reported technique realizes simultaneous single-photon denoising and super-resolution enhancement (up to 16 times), with more than 5dB superiority on PSNR compared to the existing methods.	the reported technique realizes simultaneous single-photon denoising and super-resolution enhancement (up to 4×4 times).	Redundant

Comment: It remains a bit unclear how the calibrated (multiple) noise models are efficiently added to synthetic images.

Response: Adding the multiple-noise model into synthetic images includes the following procedures.

1. We first calibrate the multiple noise model parameters. The noise formation model of dark-field images is

$$I_{dark}(x, y, n) = N_{dcr}(x, y, n) + p_{ap}(x, y)I_{dark}(x, y, n - 1) + p_{ct}(x, y)U(I_{dark}(x, y, n))$$

We could extract the noise maps of afterpulsing noise p_{ap} , and crosstalk noise p_{ct} according to their generation mechanism. The probability of afterpulsing noise is calculated as

$$p_{ap}(x, y) = \frac{\sum_n I_{ap}(x, y, n)}{\sum_n (I_{dark}(x, y, n) - I_{ap}(x, y, n) - I_{ct}(x, y, n))}$$

The probability of crosstalk noise is calculated as

$$p_{ct}(x, y) = \frac{\sum_n I_{ct}(x, y, n)}{\sum_n (I_{dark}(x, y, n) - I_{ap}(x, y, n) - I_{ct}(x, y, n))}$$

We then calibrate the dark count noise map as

$$N_{dcr}(x, y) = \frac{\sum_n (I_{\text{dark}}(x, y, n) - I_{ap}(x, y, n) - I_{ct}(x, y, n))}{n_{\text{total}}}$$

In the calibration, we mainly calibrate the parameters p_{ap}, p_{ct} and N_{dcr} .

2. We then add the calibrated noise model to synthetic images following the five steps.

(1) We create a noise-free image from public datasets and normalize it to the range $[0, \text{max_photon_number} \times \text{laser_power}]$. The max_photon_number is a typical constant corresponding to the maximum photon number of the scenes at limited photon scenarios. In this work, we use 1000 as the max_photon_number and set the laser power to 10mW, 20mW, and 40mW as the same as the experiments.

(2) For each calibrated noise source, we incorporate it into the noise-free synthetic image. Specifically, we overlay the noise model on the noise-free synthetic image on a per-pixel basis according to the calibrated parameters. In each loop, we generate a random variable R1 for each different judgment. $p(\text{Poisson})$ is the first noise probability in the noise model. If R1 is less than $p(\text{Poisson})$, we assume the pixel detected a photon at that time, so we set the pixel value to 1.

(3) We then generate another random variable R2 and compare it to the parameter p_{ap} (afterpulsing noise probability, second noise in the noise model). If the random variable R2 is less than p_{ap} and the previous frame pixel value at the same pixel is 1, it means an afterpulsing event was detected, and we set the pixel value to 1 regardless of whether there was a detected photon before.

(4) We generate a different random variable R3 and compare It with the calibrated parameter p_{ct} (crosstalk noise probability, third noise in the noise model). If the random variable R3 is less than p_{ct} and the pixel value at the neighboring pixels is 1, it means that we detected a crosstalk event. We set the pixel value to 1 regardless of whether a photon was detected before.

(5) We sum the 1-bit frames on the time axis to obtain a multi-bit grayscale image. To account for the dark count effect, we add the Poisson random map with the expected value being the dark count rate map we calibrated. This approach enables the stacking of different noise models while controlling the noise level.

Finally, we add the calibrated multiple noise model to obtain the final synthetic image with comprehensive noise characteristics from the public dataset. We summarized the above process in the following Algorithm R1, and added it in the Supplementary Note 7. Besides, we have also revised the manuscript at line 577 on page 20.

ALGORITHM R1. ADD CALIBRATED NOISED MODEL TO SYNTHETIC IMAGES

```
1 Input: a zero matrix  $I$ , the pixel index  $i$ , the time index  $t$ , a random variable  $R$  that is
generated differently each time, the  $p(\text{Poisson})$ , the neighbor region around pixel  $I(U(i,
t))$ , and probabilities  $p(\text{afterpulsing})$ , and  $p(\text{crosstalk})$  that we have calibrated, we can
define  $N_{dcr}$  as the Poisson random variable with an expectation value equal to
the .mat file we have calibrated.
2 For  $i=1, 2, \dots, N$  do
3     For  $t = 1, 2, \dots, T$  do
4         If  $R1 < p(\text{Poisson})$ 
4              $I(i, t) = 1$ 
5         If  $R2 < p(\text{afterpulsing}) \ \& \ I(i, t-1) == 1$ 
5              $I(i, t) = 1$ 
5         If  $R3 < p(\text{crosstalk}) \ \& \ U(i, t) == 1$ 
5              $I(i, t) = 1$ 
5             Summarize  $I$  in  $t$  axis
6      $S = I + D$ 
7 Return synthetic image  $S$ 
```

Comment: Apparently, the noise issue is one of the representative concerns that SPAD imaging suffers from. However, due to the use of unique electronics, the sampling/ISP scheme is also highly unique, that should be considered in synthesizing the SPAD imaging data. Any elaboration on this would be highly beneficial to strengthen the paper.

Response: Thanks for the valuable advice. To demonstrate the uniqueness of the sampling/ISP scheme of SPAD imaging, we first analyze the sampling/ISP differences between SPAD and CMOS, and then describe the strategy for simulating the sampling/ISP scheme of SPAD arrays as follows.

1. The sampling/ISP differences between SPAD and CMOS

Compared to CMOS sensors, SPAD arrays have unique sampling schemes, including direct detection mode, gated mode, and time-correlated single photon counting (TCSPC) mode. In our experiments, we employ the direct detection mode, where each pixel counts photons within a defined integration time. Due to the avalanche circuit, SPAD arrays can detect at most one photon during the integration time. In addition, noise sources in SPAD arrays differ from those in CMOS sensors, as they include signal-dependent shot noise from photon incidence, fixed-pattern noise, dark count rate, afterpulsing and crosstalk noise from electron avalanche, and deadtime noise from circuit quenching.

The ISP for CMOS sensors typically involves noise reduction, color correction,

white balance, gamma correction, and contrast enhancement. Differently, SPAD arrays generate raw data as a series of 1-bit images, with each image represented as a 64×32 matrix. In our experiments, the pixel value is either 0 or 1 at the 20ns integration time. The primary distinction between the ISP of CMOS sensors and SPAD arrays concerns the bit depth of the image. For CMOS sensors, the output image after ISP is either 8-bit or another value, depending on the analog-to-digital converter's (ADC) bit depth. For SPAD arrays, the bit depth depends on the total number of frames in the raw data.

2. Strategy for the simulation of SPAD arrays' sampling/ISP scheme

For the sampling scheme of SPAD arrays, we employ a direct detection model as

$$I(i, j, k) = f(i * \Delta x, j * \Delta y, p * \Delta t)$$

Considering the unique avalanche circuit of SPAD arrays that detects at most one photon during the integration time, we apply the Poisson distribution with probability p (Poisson)

$= 1 - p(k=0)$, where $p(k) = \frac{e^{-\lambda} \lambda^k}{k!}$. This method considers the intrinsic properties of

SPADs and their influence on image synthesis.

For the ISP scheme, raw data is represented as a series of 1-bit images, which differs from the 8-bit or 16-bit images of CMOS or CCD sensors. Therefore, when synthesizing images, we set the integration time to 20 ns, resulting in each frame pixel value being either 0 or 1. To generate an n -bit SPAD image, we accumulate 2^n continuous frames from the raw data. For the synthetic image, we follow the same process as the physics-based procedure for the n -bit image and normalize it to a range of [0,1] to be used as input of the network.

We have incorporated additional details about the unique strategy for simulating SPAD arrays' sampling/ISP scheme in the manuscript at line 493 on page 18. Moreover, we have added Supplementary Note 8 to provide a more comprehensive analysis about the differences between the sampling/ISP schemes of SPAD and CMOS, as well as the reported simulation approach.

Comment: Demonstrating the viability on multiple imaging scenarios sounds very interesting and is definitely with non-trivial effort. However, this naturally raises the concern about the domain-specific training data, or more specifically, calibrated noise models' robustness. For example, the noise characteristics in bio samples (the claimed microscope imaging application) may be highly distinguished from those 90 set scenes in the lab. Any comments and/or strategies on how to convince readers this calibrated noise model adapts to different scenarios? Domain-specific transfer-learning?

Response: First, we considered that the noise model is mainly determined by the hardware set-up of the SPAD array, instead of the experimental environment. All the multiple macroscopic and microscopic experiments we have conducted employed the same pre-trained

network, which produced satisfying enhancement performance. This validates the robustness of the reported multiple-source physical noise model and that of the designed neural network.

Second, when tackle the experiments in which the target scenes are different from that of the training dataset, one can use transfer learning to further fine-tune the pre-trained model for satisfying enhancement performance. Fine-tuning a pre-trained model requires only a small number of target scene training pairs

To demonstrate that the reported Gated Fusion Transformer network can be effectively transferred to other experiment scenarios, we conducted an experiment on the microscopic white blood cell dataset, which was not included in our synthesized training dataset. In the experiment, we first synthesized noisy white blood cell microscopic images using the calibrated noise model, and then fed them directly into the pre-trained model for image enhancement. In addition, an additional fine-tuning dataset containing 50 pairs of white blood cell images was produced to fine-tune the pre-trained model. Finally, we used the fine-tuned network to enhance the noisy white blood cell images in the testing set. We present the visualization results in Fig. R1. We can see that the reported pre-trained model is able to achieve satisfactory enhancement performance after fine-tuning with a small amount of training data.

We have added the above analysis to the Supplementary Note 9.

Figure R1: Visualized image enhancement results of simulated SPAD microscopic white blood cell with pre-trained models and fine-tuned models. The pre-trained model can achieve satisfactory SPAD image enhancement results after fine-tuning.

Comment: SPAD imaging has its unique benefits in recording ultra-fast (dynamic) scenes with the limited illumination condition, that were not well demonstrated in this version. The authors may wanna consider if this could be strengthened.

Response: Thanks for the valuable advice. To demonstrate the ultra-fast recording ability of SPAD arrays, we have further implemented two additional experiments, including the recording and enhancement of a fast-spinning fan and electrostatic discharge from a charged sphere.

Figure R2 (Figure 5 in the main text): Single-photon imaging experiment of high-speed scenarios. (a) The single-photon imaging setup used to record the rotating fan. (b) The captured single-photon images at microsecond time intervals, along with their corresponding enhanced results. (c) The single-photon imaging setup used to capture the arc. (d) The obtained single-photon images of the arc at microsecond time intervals, and their corresponding enhancement results.

The first demonstration involves recording a rapidly rotating fan. In this experiment, we captured images of the spinning fan using an experimental setup shown in Figure R2(a) (Fig. 5 in the main text). The setup consists of an objective lens, an LED light source (GCI-060401, 3W electrical power), and the SPAD arrays. The integration frame time was set to 1 microsecond. We observed that there was noticeable noise and aberration in the raw single-photon images due to heavy SPAD noise, making it difficult to distinguish the rotating blade angle of the fast fan. We applied the reported technique to enhance the recorded SPAD images. The results showed improvement compared to the initial recordings. The background appeared smoother, and the super-resolution details were further enhanced. We have measured the rotational speed to be 0.0107 rad/μs, which is approximately equivalent to 102,000 RPM (revolutions per minute).

This measured speed is consistent with that provided in the manufacturer's instructions, which state a rotating speed of about 100,000 revolutions per minute.

The second demonstration features an electrostatic ball, as presented in Fig. R2(c) (Fig.5(c)). When activated, the circuit generates a high-frequency electric field that illuminates the thin gas inside the sphere. A high voltage alternating current is applied to the sphere via a central electrode. This energy ionizes the gases, producing positively charged ions and plasma. The high voltage from the electrode then creates an arc through the plasma to the edge of the sphere, causing it to glow. We conducted recordings of the glowing process at microsecond intervals to enhance the understanding of the high-speed arc glowing process. Through observations, we noted that the duration of the arc glowing period is approximately $4\mu s$, which is consistent with Ref. [R7].

We have added a subsection titled "Experiment on High-Speed Scenarios" at line 373 on page 14 in the revised manuscript. Furthermore, we have included an experiment video in the supplementary materials that showcases high-speed scenarios recorded using SPAD arrays, as well as the enhancement results.

Comment: Shortening the main text may help.

Response: Thanks for the advice. We have revised the redundant statements in the Introduction and Conclusion sections to shorten the main text. Here is the revision table.

Table R3: Revisions for Redundant Sentences

Line	Before	After	Revised reason
135~138	We digitally synthesized the first large-scale single-photon image dataset containing 2.6 million image pairs over 17250 various scenes,	We digitally synthesized the first large-scale single-photon image dataset,	Redundant
139~141	We reported a novel gated fusion transformer network for single-photon super-resolution enhancement, achieving the state-of-the-art performance with more than 5 dB superiority on PSNR compared to the existing methods.	We reported a novel gated fusion transformer network for single-photon super-resolution enhancement, achieving the state-of-the-art performance compared to the existing methods.	Redundant
161~162	We first built an optical setup to acquire SPAD images of various macro and micro targets, as shown in Fig. 2(a).	We first built an optical setup to acquire SPAD images of various targets, as shown in Fig. 2(a).	Redundant
190~192	We calibrate the multi-source noise parameters of the collected SPAD dataset shown above. However, further super-	Despite the above collected single-photon image dataset, further super-resolution enhancement cannot be	Redundant

	resolution enhancement can't be achieved due to the lack of high-fidelity and high-resolution ground truth pairs.	achieved due to the lack of high-fidelity and high-resolution ground truth pairs.	
--	---	---	--

Response to reviewer # 2's comments

Comment: All in all, the reviewer believes that the results are convincing and could be of wide interest in the scientific community. Furthermore, the work is described with a sufficient level of detail.

Response: Thanks for the reviewer's recognition.

Comment: The concept of using synthetic SPAD data for training a super-resolution network is not new, see, e.g.:

Germán Mora-Martín, Stirling Scholes, Alice Ruget, Robert Henderson, Jonathan Leach, and Istvan Gyongy, "Video super-resolution for single-photon LIDAR," *Opt. Express* 31, 7060-7072 (2023)

Response: Thanks for the reviewer's inspiration. We have carefully studied the publication, and conducted comprehensive analysis and comparison as follows.

1. Overview of the publication.

Germán Mora-Martín et al. used synthetic depth sequences to train a 3D convolutional neural network (CNN) for denoising and upscaling (4×) depth data. They used synthetic depth sequences generated by the AirSim open-source simulator to train a 3D CNN for denoising and upscaling depth data. Then they added noise to the depth maps according to the models that are similar to Ref. R3. They introduced noise models based on Poisson and Gaussian statistics, simulating the photon timing histograms for each pixel and generating synthetic depth datasets for training with their designed CNN. Their experiments demonstrated the effectiveness of their approach using synthetic data.

2. The differences between our work and the publication.

While the reported technique shares some similarities with the publication, there are several key distinctions that set this work apart.

(1) Multi-source physical noise model

We considered a multi-source noise model based on the physics of single-photon avalanche diode (SPAD) imaging. The reported noise model incorporates signal-dependent shot noise from photon incidence, fixed-pattern noise from SPAD array's photon detection efficiency (PDE), dark count rate, afterpulsing, crosstalk noise from electron avalanche, and deadtime noise from circuit quenching shot noise. Additionally, we implemented experiment for the comparison with synthetic datasets based on Poisson and Gaussian statistics. The experiment results in Figure 2(b) demonstrate that the enhancement based on the reported noise model is superior.

(2) Calibration technique for the multi-source noise model parameters

Different from Germán Mora-Martín et al., we proposed a calibration method to calibrate the parameters in the physics-based noise model. We combined the physical

process of single-photon imaging, and first established the noise formation model of dark-field images as

$$I_{dark}(x, y, n) = N_{dcr}(x, y, n) + p_{ap}(x, y)I_{dark}(x, y, n - 1) + p_{ct}(x, y)U(I_{dark}(x, y, n)),$$

where I_{dark} denotes the detected signal at the pixel location (x, y) in the n -th frame, N_{dcr} is the dark-count rate noise map that follows Poisson distribution, p_{ap} is the fixed probability map of afterpulsing noise, p_{ct} represents the Poisson probability of crosstalk noise, and $U(I_{dark}(x, y, n))$ denotes neighbouring detected signal of pixel (x, y) .

Table R4: Contributions between Reference [23] and this Work

	Germán Mora-Martín et al.	Ours
Synthetic dataset	15500 scenes	17250 scenes, 2.6 million images in total.
Real SPAD dataset	×	90 scenes, 1022 different bit depth, 3 different illumination flux, 2790 images in total.
Multi-source physical noise model	×	Including the shot noise, crosstalk, afterpulsing, dark count and so on.
Calibration technique for the multi-source noise model parameters	×	The technique enables to calibrate parameters p_{ap} , p_{ct} and N_{dcr} in order to adapt to different hardware settings.
Network structure	3D CNN	Transformer-based network: The dense connection and gated fusion mechanisms fuse different scales of features, which helps to preserve and compensate the critical mid-high frequency image signals and enriches the local details.
Macroscopic and microscopic applications	Only macroscopic including objects at mid range (10-20 m).	Macroscopic experiments: Large-scale imaging, USAF Resolution target/ PCB board/ rotating fan/electrostatic discharging.

(3) Real SPAD dataset

To calibrate the parameters of the noise model, we collected a real SPAD dataset consisting of 64×32 -pixel images of 90 scenes, with 10 different bit depth and 3 different illumination flux. Furthermore, we performed experiments on this real dataset to validate the effectiveness of the reported technique.

(4) Transformer based neural network

Compared to existing CNN-based image enhancement models, the reported Transformer-based Gated Fusion Transformer network offers several advantages. The self-attention mechanism in the Transformer helps the network better model global features through content-based interactions between image content and attention weights, akin to spatially varying convolution. We enable long-range dependency modeling through the shifted window and Gated Fusion mechanism. Moreover, the structure of dense connection and gated fusion allows features at different scales to fully communicate and fuse, preserving critical mid-high frequency image signals during the reconstruction process and enriching local details.

(5) Various macroscopic and microscopic applications

Germán Mora-Martín et al. focused primarily on macroscopic moving scenes' depth maps. In contrast, we conducted extensive experiments on both macroscopic and microscopic scenes to validate the effectiveness of the reported technique. The experiments include large-scale imaging, imaging USAF Resolution target/PCB board, microfluidic inspection, single-photon FPM and high-speed single-photon imaging.

We have summarized the above differences in the above Table R4, and cited this publication as reference 23 in the revised manuscript, with additional description at line 67 on page 3.

Besides, we note that we have submitted this manuscript on arXiv in December last year, before the Germán Mora-Martín et al.'s publication date of February 13, 2023. The arXiv link is <https://doi.org/10.48550/arXiv.2212.13654>.

Comment: The abstract/introduction mischaracterise SPADs to some degree. It is claimed that SPADs have high levels of noise and low fill-factor. Modern SPAD have in fact very low noise floors, and close to 100% fill factor, which is why they are being developed to target low light applications: Thomas, S. Low-light imaging with SPAD pixels. Nat Electron 4, 862 (2021).

Response: Regarding the noise level, we apologize for any confusion caused by the previous statement. We aimed to highlight the influence of quantum fluctuations and various

noise sources in photon-limited situations. While it is true that SPAD arrays have low noise floors and high sensitivity to photons, it is important to note that noise sources can still have a relatively significant impact in scenarios where the number of detected photons is limited. In these cases, the noise sources may have a greater effect compared to the stable expected measurement values.

We acknowledge that SPAD arrays can detect 0 or 1 photons during the hardware integration time, leading to considerable negative effects from quantum fluctuations or missed photons. This can result in lower signal-to-noise ratios (SNR) in certain scenarios. Indeed, as quantum physics and the Law of Large Numbers dictate, generating images with more bits will reduce the impact of quantum fluctuations. We did not mean to suggest that SPADs inherently have a high noise floor, especially when compared with CMOS or CCD sensors. To avoid confusion, we have revised the noise level statements in the abstract (line 16 on page 1) and introduction (lines 45 and 78 on page 3) accordingly.

We appreciate the inspiration regarding the fill factor of modern SPAD arrays. The rapid development of SPAD technology has indeed resulted in fill factors of nearly 100%. We have cited the provided publication as reference [16], to support the revisions and acknowledge the current state of SPAD technology (line 51 on page 3).

Comment: The reviewer found the title somewhat vague; perhaps a more descriptive title could be chosen.

Response: We have updated the title to "Large-Scale High-Resolution Single-Photon Imaging," which is able to better characterize our work on improving imaging resolution and fidelity.

Specifically, to address the ambiguity surrounding the term "large-scale," we have incorporated "high-resolution" into the title. This addition emphasizes our contributions in enhancing the *optical resolution* of single-photon imaging, providing more fine details of scenes. By doing so, we aim to minimize potential confusion regarding the scalability of the hardware system. Furthermore, the inclusion of "high-resolution" highlights our efforts in denoising images and maintaining the fidelity of the acquired data. Our experiment results demonstrate that the reported technique achieves an average of 5 dB improvement on PSNR compared to existing methods.

The term "large-scale" signifies the increase in *pixel resolution* for SPAD imaging from 64×32 pixels to larger formats such as 128×64 and 256×128 pixels. As the hardware pixels expand, the reported method can generate super-resolution results. Even when the SPAD array reaches megapixel dimensions, our approach allows for increasing the imaging pixel resolution by factors of two or four in each dimension.

To sum, by updating the title to "Large-Scale High-Resolution Single-Photon

Imaging", it highlights our key contributions in improving *pixel resolution* and *optical resolution*, respectively.

Comment: The novel structural elements in the "gated fusion transformer network" used in the work should be made more explicit.

Response: Thanks for the valuable comment. The following is the detailed structure of the Gated Fusion Transformer network.

(1) Overall architecture

To achieve end-to-end SPAD image enhancement, we designed a Gated Fusion Transformer network based on the Swin Transformer structure. This network consists of three main modules: shallow feature extraction module, deep feature fusion module and image reconstruction module.

Figure R3: The detailed structure of the Gated-Fusion Transformer network

Shallow Feature Extraction: Given a low-quality image $I_{LQ} \in \mathbb{R}^{h \times w \times c_{in}}$ (h , w and c_{in} are the image's height, width and channel number), we use the shallow feature extractor $H_{FE}(\cdot)$ to explore its low-frequency features $F_0 \in \mathbb{R}^{h \times w \times c}$ as

$$F_0 = H_{FE}(I_{LQ}).$$

This module is composed of convolution, batch normalization and activation layers. The convolution layers are applied for preliminary visual processing, providing a simple way to map the input image space to a higher-dimensional feature space. Besides the following deep fusion operation, the output of this module is also linked to the image reconstruction module, so that the low-frequency information can be well preserved in final reconstruction.

Deep Feature Fusion: Next, we use several densely connected Swin-Transformer blocks (DCSTB) to extract different levels of medium-frequency and high-frequency

features $F_i \in \mathbb{R}^{h*w*c}$ ($i = 1, 2, \dots, n$) from F_0 , denoted as

$$F_i = H_{DCSTB}(F_0, F_1, \dots, F_{i-1}),$$

where H_{DCSTB} represents the i_{th} DCSTB operation. Compared to the conventional convolution blocks, the transformer blocks realize spatial-variable convolution that helps pay more attentions to the regions of fine details and interests. Consequently, such blocks help recover more high-frequency information that is beneficial to enhancing imaging resolution.

The last layer of the deep feature fusion module is the Gated Fusion layer, which fuses the outputs of different DCSTB operations with adaptively different weights. The process can be described as

$$F_{DF} = H_{GATE}(F_1, F_2, \dots, F_n) = w_1 F_1 + w_2 F_2 + \dots + w_n F_n,$$

where F_{DF} represents the multi-level deep fusion features output by the gated fusion layer, and w_n represents the weight parameters during gated fusion for different levels of feature, which are adaptively adjusted through backpropagation during network training. Such a module structure is conducive to deep mining of different levels of medium-frequency and high-frequency information, which prevents losing long-term memory as network deepens and enhances local details. The densely connected structure in deep feature fusion module helps to preserve and compensate key medium and high frequency image signals, and ensures Maximum information flow between different modules.

Image Reconstruction: We retrieve high-quality single-photon images by aggregating shallow features and multi-level deep fusion features. The operation is described as

$$I_R = H_{REC}(F_0 + F_{DF}),$$

The shallow features F_0 are mainly low-frequency image features, while the multi-level deep fusion features F_{DF} focus on recovering lost medium-frequency and high-frequency features. Benefiting from the long-term skip connections, the Gated Fusion Transformer network can effectively transmit different-frequency information to final high-quality reconstruction. Different from the state-of-the-art SwinIR network that lacks the densely connected gated fusion structure, the reported network helps preserve and compensate the key medium and high-frequency feature, and enriches image's local details. In addition, sub-pixel convolution is applied in the reconstruction block to further upsample the feature map for single-photon super resolution.

(2) *Densely Connected Swin-Transformer Block (DCSTB)*

The DCSTB module consists of K Swin-Transformer Layers (STL) and a convolutional block. Given that the input of the i_{th} DCSTB is $F_i^0 \in \mathbb{R}^{h*w*c_{in}}$ ($i = 1, 2, \dots, n$), we first use K -layers of STL to extract feature sequences $F_i^j \in$

$\mathbb{R}^{h \times w \times c}$ ($i = 1, 2, \dots, n; j = 1, 2, \dots, k$). Assuming that the input of the j_{th} STL is F_i^{j-1} , the calculation formula of its output is:

$$F_i^j = H_i^{STL_j}(F_i^{j-1}),$$

The feature sequence F_i^k processed by K-layers of STL will be input into the convolution block at the end of DCSTB module for fusion. Each DCSTB can be regarded as a short-term memory module, and the convolution block at the end adaptively gives different weights to each Swin-Transformer in the DCSTB module to fuse the output of different Swin-Transformer layers, and determines how much short-term memory the DCSTB will retain. The fused features need to be added to the input feature F_i^0 , to strengthen the long-term memory of the network. The calculation formula of the above process is,

$$F_{i,out} = H_{conv_i}(F_i^1, F_i^2, \dots, F_i^k) + F_i^0,$$

Among them, $F_{i,out}$ is the output of the i_{th} DCSTB module, and H_{conv_i} represents the convolution block in the i_{th} DCSTB module.

(3) Loss Function

We designed a hybrid loss function consisting of $L_1 - norm$ loss, perceptual loss and SSIM loss, to train the Gated Fusion Transformer network. The $L_1 - norm$ loss calculates the absolute distance between two images as $Loss_{L_1}(I_R, I_G) = \|I_R - I_G\|_{L_1}$, where I_R represents the reconstructed image by the network, and I_G denotes its ground truth. The perceptual loss is defined as the $L_2 - norm$ distance between feature maps output by the pool-3 layer of a VGG19 network pretrained on ImageNet as $Loss_{PER}(I_R, I_G) = \|\varphi(I_R) - \varphi(I_G)\|_{L_2}$, where the $\varphi(\cdot)$ operation extracts feature maps. The perceptual loss regulates different-frequency similarity in the feature space. The SSIM loss is calculated as $Loss_{SSIM} = 1 - SSIM(I_R, I_G)$, which further regulates the two images' similarity in the structural domain. To sum, the loss function for network training is,

$$Loss(I_{RHQ}, I_{HQ}) = \alpha Loss_{L_1}(I_{RHQ}, I_{HQ}) + \beta Loss_{PER}(I_{RHQ}, I_{HQ}) + \gamma Loss_{SSIM}(I_{RHQ}, I_{HQ}),$$

where α, β and γ are hyperparameters balancing the three loss parts. In this implementation, these hyperparameters were set as $\alpha = 0.1, \beta = 10$ and $\gamma = 100$ after careful network tuning.

(4) Differences with existing CNN-based models

Compared with the existing CNN-based image enhancement models, the Transformer-based Gated Fusion Transformer has several advantages. (1) The self-attention mechanism in Transformer helps the network to better model global features through content-based interactions between image content and attention weights, which

can be interpreted as spatially varying convolution. (2) Long-range dependency modelling is enabled by the shifted window and Gated Fusion mechanism. (3) The structure of dense connection and gated fusion allows features at different scales to fully communicate and fuse, which helps to preserve and compensate the critical mid-high frequency image signals in the image reconstruction process and enriches the local details.

We have added a detailed description of the Gated Fusion Transformer network to the Supplementary Note 10.

Response to reviewer # 3's comments

Comment: Overall, I think the manuscript is of high quality. The method is technically sound. The acquired data has been carefully analyzed with reasonable interpretation. These advances will move the field forward. In this regard, I believe this work has met the publication criteria of Nature Communications.

Response: Thanks for the review's recognition.

Comment: During the process of detector calibration, the presence of after-pulsing and crosstalk noise may cause the repetition of noise in the calibration, affecting the accuracy of the parameters of each other, and thereby requiring further correction of the results. I suggest the authors explain it further.

Response: Thanks for the reviewer's comment. It is indeed possible for afterpulsing and crosstalk noise to occur simultaneously, which may lead to the repetition of noise in the calibration and potentially affect the accuracy of the parameters. To address this issue, we have employed a specific counting strategy in the calibration procedure.

Specifically, based on the technical manual provided by the SPAD manufacture, the typical probability of afterpulsing events is 1-2 orders of magnitude higher than that of crosstalk events. Therefore, we prioritize afterpulsing events during the calibration process. When both afterpulsing and crosstalk events are observed at a single pixel, we classify it as an afterpulsing event rather than a crosstalk event. By doing so, we can effectively ignore the influence of crosstalk events while ensuring that afterpulsing events receive the necessary attention. This approach helps alleviate the problem of redundant calibration and enables us to obtain accurate parameter values.

We have added a section at line 560 on page 19 to highlight the above strategy.

Comment: Further clarification for the experiment of microfluidic inspection (Fig. 3) would be helpful. In particular, I would think that the micro-droplet was moving in the microfluidic channel during the image acquisition. In this case, it would lead to blur in the raw images. How would the deep transformer net handle this situation?

Response: We would like to provide further clarification on how the deep transformer network handles such situations.

First, when performing microfluidic experiments, the SPAD acquires each 8-bit grayscale image in $5\mu s$. In such a short time, the micro-droplet moves a very short distance, and the motion blur is relatively slight. Second, the resolution of the raw images acquired by SPAD is only 32×64 . That means the inherent blurring caused by the low resolution itself is much more significant than the blurring induced by motion. Consequently, there is rare motion blur in SPAD imaging, which can be generally ignored.

In the extreme case that there is motion blur, we can still tackle the degradation thanks to the transformer-based network structure. Specifically, different network layers in the Gated Fusing Transformer network can extract features at different levels, and the dense connection mechanism can make these features fully communicate and complement each other, which helps restore the local details and remove blur. The gated fusion mechanism fully integrates these features, and the fusion of different levels of features also helps to enhance and deblur the local details.

Comment: Please give the detailed structure of the feature extraction module and image reconstruction module in Fig. 5a.

Response: We summarize the comprehensive network structures of both the shallow feature extraction module and the image reconstruction module in the following Table R5. The primary function of the shallow feature extraction module is to extract shallow features from the input image while simultaneously adjusting the number of channels from 3 to 96. This process facilitates subsequent processing steps and improves the overall performance of the system. The image reconstruction module aims to enhance the image resolution and generate a high-quality output image. By leveraging advanced techniques, it effectively reconstructs the image based on the extracted features.

We have included the above details and Table R5 in Supplementary Note 3, providing a comprehensive description of the network’s structures and functionalities.

Table R5: Detailed Structure of Shallow Feature Extraction Module and Image Reconstruction Module

Feature Extraction	Output size
Conv(1,96,3,1,1)	32*32*96
Conv(96,96,3,1,1)	32*32*96
Conv(96,96,3,1,1)	32*32*96
Image Reconstruction	Output size
Conv(96,24,3,1,1)	32*32*24
LeakyRelu(24)	32*32*24
Conv(24,24,3,1,1)	32*32*24
LeakyRelu(24)	32*32*24
Conv(24,64,3,1,1)	32*32*64
LeakyRelu(64)	32*32*64
upsample(4x,64,4)	128*128*4
Conv(4,1,3,1,1) → output	128*128*1

Comment: Why add a convolutional layer at the end of each DCSTB module?

Response: Each DCSTB can be regarded as a short-term memory module. The convolution layer at the end of each DCSTB adaptively gives different weights to each Swin-Transformer to fuse the output of different Swin-Transformer layers, and determines how much short-

term memory will be retained. Specifically, there are two reasons for adding a convolution layer at the end of each DCSTB module. The first reason is that the last convolutional layer in each DCSTB module can adaptively assign a fusion weight to the residual connections from each sub-layer in DCSTB module, which is equivalent to adding a gated fusion layer at the end of each DCSTB module. The second reason is to convert 1D features into 2D features, and perform preliminary processing so that they can be fed into the final gated fusion (also implemented by the convolution layer) for weighted fusion. We have added the corresponding analysis in the Supplementary Note 10.

Comment: The figures and tables are clear and effectively illustrate the results. However, it would be helpful to provide the number of bits in the image in the figure captions, to make it clear what is being shown in each figure.

Response: Thanks for the valuable advice. We have updated the figure captions to include the number of bits in the images at line 156, 211, 214, 283, and 333.

Comment: There are many experiments presented in this work, each with different settings and specifications. It would be nice if the authors could summarize them using a supplementary table so that the readers can easily refer to them without searching for details embedded in the text.

Response: Thanks for the valuable advice. We have added a supplementary table in the Supplementary Note 11 that provides a comprehensive overview of the different experiment settings and specifications conducted in this work.

Table R6: Settings and Specifications of Different Experiments

Experiment	Figure	Light source	Image size / pixels×pixels	Enhanced size / pixels×pixels	Bit depth
Large-scale imaging	1	488nm laser power	64×32	128×64/ 256×128	1~10 bit
USAF Resolution target/ PCB board imaging	2	Olympus BX53	64×32	128×64/ 256×128	1~10 bit
Microfluidic inspection	3	Cnoptec Microscopy	64×32	128×64	1~10 bit
Single-photon FPM	4	Adafruit P4 LED	32×32	384×384	7 bit

		array			
High-speed single-photon imaging	5	GCI-060411 LED array	64×32	128×64/ 256×128	1~10 bit

For the USAF Resolution target / PCB board and Microfluidic inspection experiments, we varied the bit depth from 1 to 10 bits. Besides, we applied super-resolution techniques at the scale factors of $\times 2$ and $\times 4$ to enhance the imaging solution.

Regarding the single-photon FPM (Fourier Ptychographic Microscopy) experiments, the input images had a resolution of 32×32 pixels with an exposure time of $3 \mu\text{s}$. Through the implementation of FPM, we achieved a higher resolution of 192×192 pixels, which was subsequently further enhanced to 384×384 pixels.

In the high-speed single-photon imaging experiments, we varied the bit depth from 1 to 10 bits. Additionally, we employed super-resolution techniques at the scale factors of $\times 2$ and $\times 4$ to further enhance the results.

Comment: After the training is completed, how much time does the network consume during operation?

Response: We used RTX3090 GPU for network implementation on the python and pytorch framework. The average time to enhance a single SPAD image from 32×64 to 128×256 is 0.04s. We have added the details at line 676 on page 23.

Comment: Considering that a new neural network is a main feature of this work, the authors should make the software and these single-photon imaging datasets open source. I think many interested readers would certainly want to leverage these datasets to train their networks to test the feasibility of single-photon imaging. Thus, this action can further enhance the impact of this work.

Response: Thanks for the valuable advice. We have uploaded the demo code and data to the public repository at <https://github.com/bianlab/single-photon>.

Reference

- R1. Bolduc, E., Agnew, M. & Leach, J. Video-rate denoising of low-light-level images acquired with a SPAD camera. In 2016 Photonics North (PN), 1–1 (2016).
- R2. Chandramouli, P. et al. A bit too much? high speed imaging from sparse photon counts. In 2019 IEEE International Conference on Computational Photography (ICCP), 1–9 (2019).
- R3. A. Ruget, S. McLaughlin, R. K. Henderson, I. Gyongy, A. Halimi, and J. Leach, “Robust super-resolution depth imaging via a multi-feature fusion deep network,” *Opt. Express* 29(8), 11917–11937 (2021).
- R4. F. Zhuang et al., "A Comprehensive Survey on Transfer Learning," in *Proceedings of the IEEE*, vol. 109, no. 1, pp. 43-76, Jan. 2021, doi: 10.1109/JPROC.2020.3004555.
- R5. Germán Mora-Martín, Stirling Scholes, Alice Ruget, Robert Henderson, Jonathan Leach, and Istvan Gyongy, "Video super-resolution for single-photon LIDAR," *Opt. Express* 31, 7060-7072 (2023)
- R6. Kirillov, A., Mintun, E., Ravi, N., Mao, H., Rolland, C., Gustafson, L., Xiao, T., Whitehead, S., Berg, A.C., Lo, W.-Y., Dollár, P., & Girshick, R. (2023). Segment anything. arXiv preprint arXiv:2304.02643.
- R7. Li, J., Lei, B., Wang, J. et al. Atmospheric diffuse plasma jet formation from positive-pseudo-streamer and negative pulseless glow discharges. *Nat. Commun. Phys* 4, 64 (2021). <https://doi.org/10.1038/s42005-021-00566-8>

REVIEWERS' COMMENTS

Reviewer #1 (Remarks to the Author):

The response letter and the revised version read very detailed and in-depth, and have already resolved most of my concerns. Just one minor comment, the title, although much better than the original one, may still sound a bit ambiguous to general readers. If the key contribution is actually improving the resolution of SPAD imaging significantly, then it would not be necessary to address the "Large-scale" again. One suggestion, "High-resolution single-photon imaging with XXX" may be more appropriate and sound distinguished from such many general single-photon imaging papers. Herein, "XXX" could address the key technical insight of the method, including the gated fusion scheme, for example. Please do carefully consider this. Given a better clarified title as well as incorporating ALL those edits as mentioned/promised in the response letter, it would move towards the acceptance to the publication.

Reviewer #2 (Remarks to the Author):

The reviewer would like to thank the authors for responding and addressing the previous comments so thoroughly. The paper is recommended for publication in its current form.

Reviewer #3 (Remarks to the Author):

The authors have addressed all my comments. With the additional explanation and new data, the quality of this manuscript is also considerably improved. I believe it is ready to be published.

Response to reviewer # 1's comments

Comment: The response letter and the revised version read very detailed and in-depth, and have already resolved most of my concerns. Just one minor comment, the title, although much better than the original one, may still sound a bit ambiguous to general readers. If the key contribution is actually improving the resolution of SPAD imaging significantly, then it would not be necessary to address the "Large-scale" again. One suggestion, "High-resolution single-photon imaging with XXX" may be more appropriate and sound distinguished from such many general single-photon imaging papers. Herein, "XXX" could address the key technical insight of the method, including the gated fusion scheme, for example. Please do carefully consider this. Given a better clarified title as well as incorporating ALL those edits as mentioned/promised in the response letter, it would move towards the acceptance to the publication.

Response: Thanks for the reviewer's suggestion. We have revised the title to "High-resolution single-photon imaging with physics-informed deep learning".

Response to reviewer # 2's comments

Comment: The reviewer would like to thank the authors for responding and addressing the previous comments so thoroughly. The paper is recommended for publication in its current form.

Response: Thanks for the reviewer's recognition.

Response to reviewer # 3's comments

Comment: The authors have addressed all my comments. With the additional explanation and new data, the quality of this manuscript is also considerably improved. I believe it is ready to be published.

Response: Thanks for the review's recognition.